# Temperature-Responsive Polymer Brush Coatings for Advanced Biomedical Applications

**DOI:** 10.3390/polym14194245

**Published:** 2022-10-10

**Authors:** Svyatoslav Nastyshyn, Yuriy Stetsyshyn, Joanna Raczkowska, Yuriy Nastishin, Yuriy Melnyk, Yuriy Panchenko, Andrzej Budkowski

**Affiliations:** 1Smoluchowski Institute of Physics, Jagiellonian University, Łojasiewicza 11, 30-348 Krakow, Poland; 2Lviv Polytechnic National University, St. George’s Square 2, 79013 Lviv, Ukraine; 3Hetman Petro Sahaidachnyi National Army Academy, Heroes of Maidan St. 32, 79012 Lviv, Ukraine

**Keywords:** temperature-responsive polymers, brushes, coatings, biomedical applications

## Abstract

Modern biomedical technologies predict the application of materials and devices that not only can comply effectively with specific requirements, but also enable remote control of their functions. One of the most prospective materials for these advanced biomedical applications are materials based on temperature-responsive polymer brush coatings (TRPBCs). In this review, methods for the fabrication and characterization of TRPBCs are summarized, and possibilities for their application, as well as the advantages and disadvantages of the TRPBCs, are presented in detail. Special attention is paid to the mechanisms of thermo-responsibility of the TRPBCs. Applications of TRPBCs for temperature-switchable bacteria killing, temperature-controlled protein adsorption, cell culture, and temperature-controlled adhesion/detachment of cells and tissues are considered. The specific criteria required for the desired biomedical applications of TRPBCs are presented and discussed.

## 1. Introduction

Smart polymer brush coatings are macromolecular chains grafted at one end to a solid surface, capable of changing their physicochemical properties reversibly in response to environmental stimuli (temperature, pH, ionic pressure, light, etc.) [1], which makes them quite attractive for the health sciences [1,2,3,4]. They may be covalently bonded by one end to the surface with the other one being capable of relatively free motion. The covalent bonding of smart polymer brushes to the surface eliminates the translational freedom of macromolecular chains [5] and prevents their desorption, making them one step ahead of other surfaces in terms of durability. A similar class of materials, bottlebrush polymers, have also been actively developed in the last years [6]. Bottlebrush polymers have densely grafted side chains along the linear backbones with extended cylindrical shapes with no entanglements. Smart bottlebrush polymers can be used for the fabrication of well-organized and predictable coatings on the solid surfaces [7,8,9,10,11,12,13,14], but, in contrary to grafted polymer brushes, their long-term applications in the physiological environment and in contact with biological objects are questionable.

Especially high attention is paid to the temperature-responsive polymer brush coatings (TRPBCs), in which significant changes in biological activity may be triggered by a slight change in temperature. The commercially available product—Nunc™ Multidishes with UpCell™ Surface that allows the adhesion of the cells of the culture solution at elevated temperature and the detachment of the entire cell sheet at lower temperature is an excellent example of the application of TRPBCs [15,16]. Another example of the application of TRPBCs is a smart antibacterial surface with a kill-release strategy [17,18]. Moreover, TRPBCs enable temperature-controlled protein adsorption that is attractive for thermo-responsive chromatography and biosensing [19].

Despite progress in the development of such interesting materials, there are still some issues that need to be resolved, such as biocompatibility, high efficiency, selectivity of the action, stability, long-term and multiple use, and the temperature of the transition close to physiological temperatures (appropriate transition temperature). Therefore, there is a constant need for new approaches to design surfaces that could meet all the desired requirements.

In this review, we focus mainly on the most advanced biomedical applications of TRPBCs: temperature-switchable bacteria killing, temperature-controlled protein adsorption, and cell culture, as well as temperature-controlled detachment of the cells and tissues. Each desired application places some specific requirements for polymer coatings, and TRPBCs for advanced biomedical applications must meet the required criteria, as presented in Figure 1.

Traditional antibacterial coatings are generally constructed based on either an ‘active strategy’ to kill bacteria by using antibacterial agents or a ‘passive strategy’ to prevent bacterial adhesion with the objective of reducing the viability or number of attached bacteria [20,21]. Antibacterial agent-loaded coatings often suffer from the problems associated with the accumulation of dead bacteria, which might trigger undesired inflammation and reduce the efficiency of killing, leading to the formation of biofilms, as even a few bacteria attached to the coating can colonize quickly [20,21]. On the other hand, it remains challenging to achieve a ‘perfect’ antifouling coating with 100% bacterial prevention efficiency. According to this, the TRPBCs for temperature-switchable bacteria-killing should change their physicochemical properties from a strongly antibacterial state to a cell repellant state at the appropriate temperature, namely, to be highly effective against bacterial cells, and to have a self-cleaning effect using the switching on/off of the temperature, with an appropriate transition temperature, and long-term application including multiple cycles of temperature-induced transitions. Moreover, such coatings should be nontoxic toward eukaryotic cells, which is a very challenging issue, as usually the mechanisms that are responsible for the death of bacteria cells are effective also toward other cell types.

Advanced applications for temperature-controlled protein adsorption can be divided into two subcategories: temperature-controlled protein fouling/non-fouling or protein separation [22,23,24] and temperature-controlled protein orientation [25]. Controlling the interaction between surfaces and proteins in an aqueous environment is important for the prediction of the protein adsorption during the first step of surface fouling. Moreover, in recent years, temperature-controlled protein separation turned out to be a powerful method for separating a given protein from a protein mixture. The TRPBCs for advanced application as the temperature-controlled protein fouling/non-fouling materials should undergo the transition from a highly adsorbing state to a strongly non-fouling state in response to small changes in the temperature in the physiological environment. In addition, TRPBCs should be able to separate a single protein from the mixture of the proteins and trigger very rapid adsorption/desorption of the protein in response to small changes in temperature. Another approach is the advanced application of TRPBCs for temperature-controlled protein orientation. The orientation of proteins on the surface plays a crucial role in the binding of cell receptors and ultimately defines their targeting efficiency [26]. TRPBCs for temperature-controlled protein orientation, first, should not cause denaturation of the adsorbed proteins, and second, the temperature of the transition should be close to physiological (appropriate temperature).

The damaged tissue of the patient can hypothetically be recovered by introducing the external cells or by implantation of the external tissue. However, the application of external cells to restore damaged tissue requires a long incubation time, whereas many of the introduced cells may lose their functions in a much shorter time. Therefore, the introduction of external tissues is a promising strategy for the restoration of damaged tissues. The most suitable approach for tissue engineering is cell sheet engineering employing TRPBCs. The main requirements for TRPBCs for advanced cell culturing and detachment are the following: perfect biocompatibility, high adhesion, proliferation and viability of the cells on the coatings, controlled and rapid cell or tissue detachment, appropriate transition temperature, and the ability for separation of the different cellular types. Application of the TRPBCs for temperature-modulated separation of the different cellular types is based on the adhesion and detachment of different types of cells on the same coatings, which is enabled by the change of physicochemical properties at different temperatures. This separation method is very important for various applications in biotechnology and biomedicine, because it is possible to not use reagents that damage and deactivate biological compounds, proteins, and cells.

Cell adhesion and release, bacteria attachment, and protein adsorption are usually governed by nonspecific physical interactions with TRPBCs. Nonspecific physical interactions include van der Waals, electrostatic, steric, hydration, and hydrophobic forces. In some cases, specific biological active motifs can be inserted into TRPBCs, which essentially improves the efficiency of these systems. In the work [27], copolymer TRPBC, including *N*-isopropylacrylamide (*N*IPAM) as temperature responsive units and 2-lactobionamidoethyl methacrylate as specific biological active motifs, was synthesized. In turn, in the works [28,29], copolymer TRPBCs of poly(*N*IPAM)-*block*-poly(acrylic acid) and poly(2-(2-methoxyethoxy)ethyl methacrylate) with RGD motifs responsible for the acceleration of cell attachment were synthesized. Motivated by the strong progress in TRPBCs, in this paper we made an attempt to systematize the recent advances in the engineering of advanced biomedical TRPBCs. Additionally, the mechanisms of the temperature-induced transition of TRPBCs and methods of their fabrication and characterization were discussed in detail. Finally, three groups of advanced biomedical applications of the TRPBCs: temperature-stimulated bacteria killing, temperature-controlled proteins adsorption, as well as culturing and temperature-controlled detachment of the cells, were complexly described. The crucial requirements for TRPBCs suitable for advanced biomedical applications have been analyzed and listed.

## 2. Mechanisms of the Temperature-Induced Transition of TRPBCs

The TRPBCs exhibit the response to temperature governed by different mechanisms attributed to intermolecular interactions of the macromolecular chains between themselves and with the environment [1,25,30,31]. The mechanism responsible for the temperature-dependent properties of polymer brushes is strongly dependent on the chemical nature of the macromolecular chains. 

The first group of transitions is related to critical solution temperatures (Upper and Lower Critical Solution Temperatures (UCST and LCST)) and cannot be realized without surrounding solvents [32,33,34]. For advanced biomedical applications, TRPBSs with LCST based on *N*IPAM, oligo(ethylene glycol)ethyl ether methacrylate with Mw = 246 (OEGMA246), di(ethylene glycol)methyl ether methacrylate (other names 2-(2-methoxyethoxy)ethyl methacrylate or OEGMA188) (DEGMA) and their copolymers with various monomers are manufactured and exploited [1].

The second group of transitions is related to temperature-induced changes in the physical state of polymers, where the presence of solvent is not required (glass–glass, glass-rubber (Tg), nematic-isotropic temperature transitions in polymers) [35]. In this paper, we focus on transitions based on LCST and Tg, which are most suitable for fabrication of the biomedical devices. These transitions are schematically sketched in Figure 2.

### 2.1. TRPBCs with Critical Solution Temperatures

The LCST (UCST) is the critical temperature below (above) which the components of a mixture are miscible for all concentrations. Transitions governed by LCST (and UCST) of polymers result in a change in the thickness and wettability of polymer brush coatings [1,36]. Below LCST, macromolecular chains are in an extended hydrophilic chain conformation, miscible with the environment [37]. When the temperature increases above the LCST, the macromolecules collapse and transform into collapsed hydrophobic globules weakly miscible with the environment [37]. This scenario is opposite for polymers with UCST, where the macromolecular chains are in the collapsed hydrophobic state below UCST and in the extended hydrophilic chain conformation above UCST [33,38,39,40,41,42]. In polymer chemistry, the phenomenon of the LCST is related to the systems based on polymer-solvent mixtures that are miscible below a given critical temperature and turn to two-phase unmixed systems above this critical temperature. The Gibbs free energy change (ΔG) related to the mixing of these two phases is negative below the LCST and positive above it, and the entropy change ΔS = −(dΔG/dT) is negative for the mixing process. This is in contrast to the more common and intuitive case, in which the entropy change promotes mixing because of the increased volume accessible to each component upon mixing. In general, the unfavorable entropy of mixing responsible for the LCST may have two physical origins. The first is related to interactions between the two components, such as strong polar interactions or hydrogen bonds, which prevent random mixing. The second physical factor that can lead to LCST is compressibility effects, especially in polymer-solvent systems [43,44]. In contrast to the LCST, the UCST is the critical temperature above which the components of a mixture are miscible in all proportions. Phase separation at the UCST is driven by unfavorable energetics; in particular, interactions between components favor a partially demixed state [43,44].

The TRPBCs most commonly used in biomedicine include OEGMA- or *N*IPAM-based polymer systems. The conformations of the macromolecular chains for these polymers are governed by hydrogen bonding, as sketched in Figure 3. For P*N*IPAM and copolymers with *N*IPAM fragments, hydrogen bonds between hydrophilic amide groups of *N*IPAM and water are established at T < LCST. Once the temperature increases above LCST, these bonds break, and other hydrogen bonds are established between the amide groups in the *N*IPAM chains [1,45,46,47] (Figure 3a). In case the of POEGMA, hydrogen bonds occur between the ether oxygen of poly(ethylene glycol) and water hydrogens at T < LCST, while at T > LCST the hydrogen bonds between the ethers in polymer chains are dominant (Figure 3b) [47,48,49,50]. Transitions are accompanied by changes in the volume and surface hydrophilicity. In the work [51], a series of dense water-swollen polymer brushes was studied using contact angle measurements, ellipsometry and quartz crystal microbalance. Diagrams of surface versus volume hydrophilicity of the brushes allow one to identify two types of behavior: strongly water-swollen brushes exhibit a progressive decrease in volume hydrophilicity with temperature, while surface hydrophilicity changes moderately; weakly water-swollen brushes have a close-to-constant volume hydrophilicity, while surface hydrophilicity decreases with temperature. Thermoresponsive brushes abruptly switch from one behavior to the other and do not exhibit an abrupt change of surface hydrophilicity throughout their collapse transition. In general, there is no direct correlation between surface hydrophilicity and volume hydrophilicity, because surface properties depend on the details of conformation and composition at the surface, while volume properties are averaged over a finite region within the brush [51]. In contrast to results reported in the work [51], our works [1,47,48,49] always showed strong changes in surface hydrophilicity at temperature-induced transitions. These differences may be related to the different structures of TRPBCs (thickness, grafting density, or other factors).

Similar mechanisms of the temperature-induced response for LCST-based systems were shown for TRPBCs based on *N*,*N*-dimethylaminoethyl methacrylate, 4-vinyl pyridine, derivatives of amino acids and polypeptides, and pentaerythritolmonomethacrylate [52,53,54,55,56].

Poly(*N*-acryloylglycinamide) (P*N*AGA) is the most known polymer with UCST [33,41,57]. In the previous works, it was shown that at T < UCST the hydrogen bonds between the carbonyl and amine groups of the P*N*AGA were observed. They were broken at T > UCST where the interaction with water prevailed for both groups (see Figure 3c). 

The crucial parameters determining the properties of the thermoresponsive grafted polymer brushes include polymer density, molecular weight, topology, and chemical component dissolved in the solvent or included in the structure of the macromolecules. The behavior of poly(*N*-isopropylacrylamide) (P*N*IPAM) grafted brushes at low grafting densities and molecular weights, as well as at high grafting densities and molecular weights, was described by Leckband et al. [58]. At low densities of grafting and molecular weights, the formation of lateral aggregates or “octopus micelles” was demonstrated. In contrast, at high grafting densities and molecular weights, the P*N*IPAM-grafted brush coatings collapsed uniformly. In turn, Benetti [59,60] showed the impact of macromolecular topology (linear versus cyclic polymer brushes) on colloidal stability and bio-inertness as well as temperature responsivity. It was noted that the properties and behavior of the examined polymer brushes were significantly different, even at the same temperature. 

Another interesting work [61] reported the synthesis and detailed characterization of thermoresponsive poly(*N*,*N*-dimethylaminoethyl methacrylate) TRPBC with well-controlled molecular weight and grafting density (0.08–0.20 chains/nm^2^). For this material, a well-pronounced LCST transition is observed with a reduction in brush layer thickness of more than 40% by spectroscopic ellipsometry at intermediate grafting densities (0.12–0.20 chains/nm^2^) in 5 mM NaCl solution. In turn, the UCST transition, induced by multivalent [Fe(CN)_6_]^3−^ ions, reaches a remarkable change in layer thickness of ~80% already at the lowest investigated grafting density of 0.08 chains/nm^2^.

A chemical component dissolved in the solvent or included in the structure of the macromolecules has an important impact on the temperature-responsive properties of the TRPBCs. In the works [1,47,48], carboxylic groups in the structure of the multifunctional initiator, which initiates the grafting from the surface, blocked the temperature-responsive properties of the P*N*IPAM or POEGMA-based grafted brushes at acidic pH. Recently, it was shown [49,62,63] that the LCST value of POEGMA or P*N*IPAM TRPBCs in buffer differs from that determined in water because the ions of the buffer contribute to the interactions between the macromolecules themselves as well as between macromolecules and the buffer. Moreover, additional components introduced into TRPBCs may influence their LCST. Incorporation of silver nanoparticles in POEGMA brushes leads to a decrease in LCST from 29.7 to 21.6 °C, while at a high concentration of silver nanoparticles the temperature-responsive properties of POEGMA-based TRPBCs were blocked [64].

### 2.2. TRPBCs with Tg

The glass-rubber transitions of the polymers or α-relaxation influence the elasticity of the polymeric surface [25,31,35,65,66,67]. Below the glass-rubber transition temperature, the polymer is in the hard glassy state; above the glass-rubber transition temperature, the polymer is in the soft rubbery state. In the glassy state, the neighboring macromolecules interact quite strongly, tending to link together. Above the glass-rubber transition, the neighboring macromolecules interact weaker. This also affects the morphology of the polymer grafted brush coatings, which are strongly transformed from highly rough and structured at T < Tg to relatively smooth at T > Tg.

It should be noted that temperature-induced transitions, such as glass–glass transitions or β-relaxation might have a weak influence on surface elasticity while showing a considerable influence on wettability, heat capacity, and refractive index [35,66,67]. Glass–glass transitions can be attributed to a particular molecular rearrangement in the glassy polymer state, significantly less expressed than in the glass-rubber transition. Liquid crystalline polymers form orientationally ordered anisotropic liquid crystalline phases in a well-defined temperature range [35,66,67,68,69]. Polymers that undergo these transitions are used for temperature-controlled orientation of proteins [25], aligning of liquid crystals [66], and are promising materials for temperature-stimulated cell detachment [31,67].

The glass transition was reported for poly(butyl methacrylate) (PBMA) TRPBCs [31], where it is shown that Tg is depends on the thickness of the brush coating. The dependence of transition temperature on the thickness of the coatings was also demonstrated for poly(methyl methacrylate) and polystyrene TRPBCs [70,71,72].

## 3. Fabrication of TRPBC Coatings

The TRPBCs may be grafted onto the surface by either physisorption or chemisorption. TRPBCs grafted via physisorption are not stable and may desorb from the solid even at thermodynamic equilibrium. Only chemisorption allows for the achievement of TRPBC strongly bonded to the surface. There are two types of chemisorption of brushes on the surface: “grafting from” and “grafting to”. “Grafting from” implies in situ polymerization of the monomer directly from the surface, whereas the “grafting to” method involves the synthesis of a polymer with a reactive end group, followed by attachment to the surface. For the “grafting from” approach, the surface is premodified by a multifunctional initiator, while for the “grafting onto”, both macromolecules and the surface are premodified by complementary end groups capable of covalent binding. The “grafting from” method provides a high grafting density and a high molecular weight of molecular chains; therefore, we focus on this method. In the work [1], six major surface-initiated radical polymerization techniques using multifunctional initiators were noticed, including surface-initiated atom transfer radical polymerization (SI-ATRP), surface reversible addition fragmentation chain transfer polymerization (SI-RAFT), surface-initiated nitroxide-mediated polymerization (SI-NMP), surface-initiated photoiniferter-mediated polymerization (SI-PIMP), surface-initiated photopolymerization (SIPhotoP) and surface-initiated polymerization using peroxide initiators or azo initiators (SI-PP or SI-AP).

The multifunctional initiators of the grafting “from surface” are chemicals containing at least two functional groups: the group that links the surface to which the TRPBCs are grafted, and the group that initiates the growth of the macromolecular chains. To achieve highly monodisperse polymer brushes, the controlled living polymerizations are applied. During controlled polymerization, the growing molecular chain switch from active to dormant states. Two methods of controlled living grafting from polymerization are most popular nowadays: SI-ATRP and SI-RAFT.

In addition, surface-initiated ring-opening metathesis polymerization (SI-ROMP) was used as a fundamental technology for the preparation of polymer brushes. SI-ROMP offers an effective way of polymerization of norbornene monomers, allowing the preparation of specific polymers that are not accessible by other polymerization methods [73,74].

Figure 4 presents: schematically functionalization of surfaces (a), subsequent grafting of the multifunctional initiator for controlled living surface-initiated polymerization (b)**,** polymerization of the reactive monomer, initiated by reactive groups of the multifunctional initiator (c)**,** and resulting in grafted polymer brushes (d).

One of the first attempts to synthesize grafted polymer brushes using multifunctional ATRP (atom transfer radical polymerization) initiators was described in [75], where grafted polyacrylamide macromolecules were synthesized by polymerization from the surface of silica prefunctionalized with benzyl chloride. Later, Fukuda et al. [76] used immobilized 2-(4-chlorosulfonylphenyl)ethylsilane to synthesize grafted poly(methyl methacrylate) brushes. The grafting from the surface using ATRP strategy was intensively developed by K. Matyjaszewski with colleagues, who synthesized grafted polystyrene brushes on silicone surfaces premodified with 2-bromoisobutyrate residues [77].

Five components are required for SI-ATRP [78,79,80]: initiator (multifunctional alkyl halide), metallic catalyst (containing the transition metal and halogen), ligand, solvent and monomer. Polymerization contains three steps: initiation, propagation, and termination. At the initiation step, the initiator generates the radical. The transition metal from metallic catalysis during the ATRP is oxidized from a lower to higher oxidation state and takes on the halogen that originally was the part of the initiator creating the radical. Once this radical is formed, it can initiate the growth of a polymer chain and polymerize the monomer, forming an active polymer chain. During the ATRP the growing chain transforms to the dormant state when the halogen of the metallic catalyst links to it. Once the transition metal takes on the halogen from the growing molecular chain, it transforms to the active state. Typical multifunctional initiators for SI-ATRP are presented in Figure 5.

Another common method for the fabrication of the grafted brushes is RAFT (reversible addition fragmentation chain transfer polymerization) [81,82], which allows the grafting polymerization of a wide diversity of monomers. The first component of RAFT is the initiator. The typical multifunctional initiators for SI-RAFT are presented in Figure 6. The second component is the so-called RAFT agent belonging to thiocarbonylthio compounds such as dithioesters, dithiocarbamates, trithiocarbonates, and xanthates, containing two groups: R-group and Z-group [83]. Attachment of the RAFT agent through the Z-group or the R-group can considerably influence the outcome of polymer brush grafting [84,85]. As a rule, the R-group initiates the growth of the majority of polymer chains, and the Z-group stabilizes the intermediate radical species. In this case, R-designed RAFT agents allow the termination of two macroradicals on the surface, resulting in the loss of RAFT agent. In contrast, Z-designed RAFT agents prevent these side reactions. However, the transfer of the macroradical to the RAFT agent takes place close to the surface. With increasing brush length, the thiocarbonylthio group may become less and less accessible and the growing polymer brush has a shielding effect [84,85].

The third component is the solvent, which is optional, and the fourth is a monomer. The RAFT polymerization contains six steps: initiation, propagation, RAFT pre-equilibrium, reinitiation, main RAFT equilibrium, and termination. At the initiation step, the initiator is decomposed, and the radical is formed. Then, at the propagation step, the polymerization propagates. At the RAFT pre-equilibrium step, the active polymer chain reacts with the RAFT-agent to form the RAFT adduct radical and the polymeric chain containing a part of the RAFT-agent. Then, at the reinitiation step, this adduct radical initiates the growth of new polymer chains. The main RAFT equilibrium step is the most important step in RAFT polymerization for a uniform distribution of the polymer chains. At the main RAFT equilibrium step, the growing polymeric chain initiated by the RAFT adduct radical links to the polymeric chain containing the part of the RAFT agent, and the RAFT intermediate radical species are formed. Next, the part of the polymer from another side of the intermediate radical species detaches and undergoes further polymerization. When the polymeric chain contains the part of the RAFT agent, it is in the dormant state; once the chain detaches from the part of the RAFT it transforms to the active state. The termination may occur when the monomer is finished, or two active chains link together, or when disproportionation happens.

Special attention should be paid to the more progressive type of SI-RAFT named as surface-initiated photoinduced electron transfer-reversible addition–fragmentation chain transfer polymerization (SI-PET-RAFT). This method allows surface functionalization with spatiotemporal control and provides oxygen tolerance under ambient conditions [86,87]. The modularity and versatility of SI-PET-RAFT are highlighted through significant flexibility with respect to the choice of monomer, light source and wavelength, and photoredox catalyst. The ability to obtain complex patterns in the presence of air is a significant advantage compared to other controllable surface-initiated polymerization methods [86,87].

The ATRP and RAFT polymerizations contain toxic agents such as heavy metals or thiocarbonylthio compounds.

Another way of manufacturing TRPBCs involves non-controllable surface-initiated polymerization using multifunctional peroxide initiators or azo initiators [88,89,90,91]. Sometimes, the application of the coatings does not require high monodispersity of the brush, and non-controllable polymerization may be employed as well. The most common multifunctional initiators for SI-PP or SI-AP are presented in Figure 7.

In Table 1 examples of the most common multifunctional initiators for surface-initiated polymerizations are summarized.

The TRPBCs with LCST that are most commonly used for advanced biomedical applications are fabricated from *N*-isopropylacrylamide (*N*IPAM) (1), oligo(ethylene glycol)ethyl ether methacrylate with Mw = 246 (OEGMA246) (2), di(ethylene glycol)methyl ether methacrylate (DEGMA or OEGMA188) (3), *N*,*N*-dimethylaminoethyl methacrylate (4) and their copolymers with units from other functional monomers (5–8) (see Figure 8) [1]. Recently, increasing attention has been paid to TRPBCs with UCST, especially poly(*N*-acryloyl glycinamide-*co*-*N*-phenylacrylamide) (9) or poly(imidazoled glycidyl methacrylate-*co*-diethylene glycol methyl ether methacrylate) (10). Additionally, we demonstrated TRPBCs with Tg close to physiological conditions based on poly(butyl methacrylate) (PBMA) (11) and poly(cholesteryl methacrylate) (12)**,** which are promising for biomedical applications. Other chemical structures of the TRPBCs will be noted below in sections related to their advanced biomedical applications.

## 4. Methods for the Determination of Transitions in TRPBC Coatings

Depending on the application, the TRPBCs can be immobilized onto different surfaces. The TRPBCs may be attached to flat planar surfaces (e.g., glass plates or silicon wafers) for tissue engineering, alignment of liquid crystals, and the killing of bacteria on daily-used frequently touched surfaces. If the application requires a large surface area or the ability to disperse the samples in blood (for drug or gene delivery) or other solvents (for temperature-responsive chromatography), the TRPBCs may be grafted to dispersive objects such as nanoparticles, microparticles, nanotubes, and many others. In general, the different methods are suitable to characterize grafted brushes immobilized onto flat and curved surfaces or dispersive materials. The main methods enabling the characterization of the TRPBCs are presented in Table 2.

The LCTS-based transition of TRPBCs appears as a drastic change in wettability [1,47,48,92,93,94]. To determine the temperature of the transition between the states of the extended hydrophobic chain and the collapsed hydrophilic globule, the contact angles of water at different temperatures are often studied [1,47,48,92,93,94]. The temperature dependence of the water contact angle is similar to that of a sigmoid with the transition temperature at its inflection point (middle point) [1,47,48,92,93,94]. Stetsyshyn et al. studied the influence of glass transition temperature on the wettability of poly(butyl methacrylate) TRPBCs and established that above the glass transition temperature the surface is more hydrophobic than below, but the shape of the curve showing the temperature change in wettability strongly depends on the thickness of the brush and does not allow us to determine Tg [25,31]. The measurements of the contact angles are not applicable for brushes that are immobilized on the dispersive surfaces. In fact, drop shape visualization and contact angle measurement are also possible on curved surfaces [95] but these measurements were not performed for TRPBCs [25,31]. 

Optical methods such as ellipsometry and white light reflectance spectroscopy (WLRS) and others are suitable for characterization of the TRPBCs on flat surfaces. Once the temperature increases above LCST, the TRPBC collapses and the thickness drops quite sharply, which is detectable with ellipsometry and WLRS [96]. Varma et al. reported that the swelling ratio of P*N*IPAM above LCST is twice as small as that above LCST [97]. In the work [98], an original ellipsometric technique was developed and used to independently determine the strongly correlated refractive index and thickness of transparent ultrathin TRPBC films. The glass transitions of TRPBCs detectably influence their thermal expansion coefficient [70,71,72]. Zuo et al. recorded the thermal dependence of the normalized thickness of polystyrene TRPBCs around their glass transition temperature and reported the Tg values defined by the deflection point on the corresponding curves [71].

While ellipsometry and WLRS are appropriate techniques to study flat surfaces, for the case of dispersive nanosized TRPBCs, dynamic light scattering (DLS) is generally used [64,99,100,101,102,103]. Usually, the thickness of the polymer brushes that are attached to nanosized objects is too low for the study of the thermal expansion of the polymer brush coating; hence, it is difficult to find the glass transition temperature of dispersive TRPBCs with DLS. The DLS allows for the determination of the ξ-potential and hydrodynamic radius of the TRPBCs immobilized on dispersive objects [64,99,100,101,102,103,104]. It is worth noticing that in the phosphate buffers, the TRPBCs may be saturated by ions, and consequently, the LCST will not result in a measurable hydrodynamic radius.

In the work [105], it is reported that DSC is applicable to determine LCST transitions for thermoresponsive nanoparticles with polymer grafted shells in water. The LCST appears in the DSC thermograms as an endothermic peak. The glass transition temperature of dispersive TRPBCs is quite often determined by the DSC [106,107]. The DSC measures as a function of temperature the difference in heat flux needed to increase the temperature of the sample and the reference. The glass transition of the polymer brush coating results in a smooth step in the heat capacity of the polymer brush. L. Zhu et al. recorded the DSC heating curves of polystyrene brushes grafted to Au nanoparticles and have shown that the curves contain the peak and deflection at the glass transition temperature [108]. 

LCST affects the surface morphology of TRPBCs, but the influence is complicated, and therefore it is practically impossible to determine the LCST with AFM studies [49]. Contrary to LCST, Tg of TRPBCs can be easily determined using AFM [25,31].

The LCST transition of polymer brush chains affects their miscibility with the environment [109,110]. Below LCST, the polymer brush coating is well miscible with the environment and almost transparent; once the temperature is increased above the LCST, the polymer brush becomes badly miscible with the environment, starts to aggregate, and the system becomes turbid. The temperature dependence of the absorbance around the LCST is similar to that of the sigmoid with the inflection point at the LCST. The method of turbidity measurements is applicable for the polymer brushes attached to the dispersive objects and to the polymer brushes attached to the flat bulk surfaces [109,110].

Finally, to determine the biological activity of TRPBCs with respect to cells, bacteria, and proteins, techniques and methods specific for each of these biomedical applications (cell cultures, antibacterial tests, protein adsorption, and binding assays) are usually used (not discussed here). Instead, we mention isothermal titration calorimetry (ITC), a remarkable technique that allows us to study the interactions between the proteins and the nanoparticles modified by grafted polymer brushes. Recently, ITC has been applied to examine the adsorption of different proteins on TRPBC-functionalized nanoparticles, both below and above the LCST of polymer brushes [104]. ITC provides thermodynamic parameters for the interactions, including free energy ΔG, entropy ΔS and enthalpy ΔH changes for protein adsorption [104,111,112]. This set of parameters is often not complete to unequivocally determine the mechanism of association [111,112]. In the work [104], both nanoparticles and proteins had negative ξ-potentials and therefore only two types of associations were allowed: hydrophobic or polar hydrophilic. The hydrophobic nanoparticles are surrounded by “secondary bound water” that performs hydrophobic hydration around the side chain carbon atoms, where cage-like water formations around these carbons allow the polymer to remain in the water [49,104]. When the water from hydrophobic hydration is released, it recombines, and the association is exothermic (with ΔH < 0) and driven by enthalpy. In turn, a hydrophilic nanoparticle is surrounded by primary water, which forms hydrogen bonds with its surface. When primary water is released (resulting in ΔS > 0), heat is absorbed, but the primary water released later establishes the H-bonds with the bulk water, and heat is released. Only the case of positive enthalpy change ΔH and entropy gain ΔS can be unequivocally assigned to the polar hydrophilic association due to entropy-driven interactions. For the proteins examined in [104], the mechanism of their adsorption on TRPBC-functionalized nanoparticles was observed to change from entropic to enthalpic with the temperature elevated above the LCST of the brushes. 

## 5. Advanced Biomedical Applications of TRPBCs

### 5.1. Antibacterial TRPBCs

The antibacterial surfaces that are known nowadays are divided into two groups: (I) passive protection coatings where nonfouling surfaces prevent the adhesion of dangerous bacteria and (II) active protection coatings where the surfaces containing antibacterial agents, such as antibiotics, kill the bacteria [17,18,113]. Surfaces that are antifouling for bacteria often can also be antifouling against some proteins and cells, and hence have limited applications. High interest is paid to switchable surfaces that show their antibacterial activity in response to external stimuli (e.g., light, temperature, pH). Stimuli-responsive coatings have at least two advantages: first, antibacterial properties can be remotely switched on in the necessary time and second, combination of the antifouling and “smart” properties of the coatings can lead to enable from the cleaning away of dead bacteria or proteins using the switching on/off method. Zwitterionic polymer-based coatings, such as poly(carboxybetaine), poly(sulfobetaine) and poly(phosphataine) containing positively and negatively charged units, are one of the most widely used antibacterial coatings [18]. High attention is also paid to the engineering of antibacterial surfaces that are potentially stimulated by light irradiation [114]. However, TRPBCs remain the most interesting antibacterial materials mainly because they are easy for both, fabrication and applications.

Passive temperature-stimulated anti-bacterial coatings based on P*N*IPAM exhibit the temperature-controllable wettability driven by LSCT that results in controlled adhesion or detachment of bacterial cells [115]. This process is strongly dependent on the interactions between the given bacteria strain and TRPBC. For example, *Cobetia marina*, attached well to P*N*IPAM grafted brush coatings at T > LCST is easily released from the surface at T < LCST [116,117,118]. On the contrary, *Staphylococcus epidermidis*, attached to P*N*IPAM coatings in a high amount at T < LCST may be rinsed at T > LCST [117]. In the work [119], the attachment of *Salmonella typhimurium* and *Bacillus cereus* was tested for brushes based on P*N*IPAM brushes and its copolymers. Strong adhesion of *Salmonella typhimurium*, as well as *Bacillus cereus*, to P*N*IPAM coatings was observed at T > LCST and weak adhesion at T < LCST. In turn, for poly(*N*IPAM-*co*-acrylamide) (85 to 15 mol%) copolymer grafted brush coatings, almost the same adhesion was shown at different temperatures for each type of bacteria. In contrast, for poly(*N*IPAM-*co*-*N*-tert-butylacrylamide) (80 to 20 mol%) very strong adhesion at T > LCST and very weak adhesion at T < LCST was observed for both bacteria [119].

The antibacterial activity of self-disinfected “active” TRPBCs can be achieved by the action of the antibacterial fragments in the structure of the polymer coatings or by the release of the antibacterial substances previously immobilized in the TRPBCs. This type of TRPBCs also has self-cleaning properties at temperatures below LCST, where the polymer becomes hydrophilic and consequently often releases dead bacteria [120,121,122,123,124,125]. The information on antibacterial TRPBC systems is summarized in Table 3.

In the work [120], a system based on temperature-responsive PDEGMA brushes was developed on titanium implants for a controlled and thermally triggered release of the antibiotic levofloxacin at the wound site. Levofloxacin release at T > LCST was provided. The antifouling effects of the PDEGMA coating additionally enhanced the bactericidal effects.

In the works [92,93], temperature-stimulated antibacterial coatings based on PDEGMA, and noncytotoxic to human cells, were developed. These TRPBCs exhibit the LCST at a temperature appropriate for biomedical applications, and their antibacterial activity was provided by the silver nanoparticles incorporated into the brush. The antibacterial activity of the coating was switched on by elevating the temperature above the LCTS, activating the release of silver ions. Additionally, contact killing was possible by direct interaction of bacterial cells and silver nanoparticles, facing the surface of the coating at the collapsed state of the grafted brushes. Antibacterial activity was not observed at T < LCST. Furthermore, the gradual release of silver from PDEGMA-based TRPBCs reported in the work [92] ensures the possibility of long-term use and the durability of the antibacterial activity, which is an important issue in the application of antibacterial coatings. A similar effect was demonstrated for poly(4-vinylpyridine) TRPBCs with embedded silver nanoparticles [93].

In the works [121,122], copolymers of *N*IPAM and bactericidal [2-(methacryloyloxy)ethyl]trimethylammonium chloride attached to the surface were synthesized. Quaternary ammonium salts effectively killed the attached bacteria (*E. coli* and *S. aureus*) at T > LCST and the dead bacteria were detached by reducing the temperature below the LCST. No detachment was observed at T > LCST.

Another approach was used in the work [123], in which mixed grafted brushes of P*N*IPAM and poly [2-(methacryloyloxy)ethyl]trimethylammonium chloride were synthesized. Above the LCST, the P*N*IPAM chains collapsed to expose poly [2-(methacryloyloxy)ethyl]trimethylammonium chloride chains which were able to kill attached bacteria cells. Below the LCST, the P*N*IPAM chains became more hydrophilic, leading to the detachment of bacterial debris.

Temperature-responsive antibacterial coatings based on poly(*N*IPAM-*co*-2-carboxyethyl acrylate) copolymer and vancomycin moieties were built on a poly(sulfobetaine methacrylate) surface [124]. At T < LCST, the TRPBC kills the bacteria by vancomycin moieties. At T > LCST, the TRPBC collapsed, showing notable performances in bacterial inhibition and dead bacteria detachment.

In the work [125], the composition of the temperature-responsive copolymer brushes based on DEGMA, hydroxyl-terminated oligo(ethylene glycol) methacrylate, and 2-hydroxyethyl methacrylate was adjusted to obtain a collapse temperature of ~35 °C. Grafted brushes were modified with antibacterial magainin I peptide, whose activity was tested at different temperatures against *L. ivanovii* and *E. coli*. The surface properties of the peptide-functionalized brushes have changed from dominantly bactericidal at 26 °C to dominantly non-adhesive when the temperature becomes slightly higher than that of LCST.

The mechanism of the kill–release strategy that includes the temperature-induced release of antibacterial agents previously immobilized in TRPBCs followed by the temperature-induced self-cleaning of the coatings is sketched in Figure 9. Below the LCST, the TRPBCs are surrounded by the hydration layer and the bacteria do not adhere to the surface of the TRPBCs. Once the temperature is increased above the LCST, three ways of antibacterial activity are possible. In the first way, bacteria adhere to the TRPBCs and are killed by the embedded antibacterial agent. In the second, the TRPBCs begin to release the antibacterial agent and kill the non-adhered bacteria. The third way combines both the mechanisms described above the LCST. The dead bacteria adhered to the brush are released when the temperature of the brush is decreased below the LCST.

Advanced antibacterial systems based on TRPBCs must meet some criteria to be effective for application. They must be highly effective against bacterial cells and nontoxic to eukaryotic cells, present the self-cleaning effect using on/off switching by the temperature, have appropriate transition temperature, and provide long-term application including multiple cycles of temperature-induced transitions.

### 5.2. TRPBCs for Cell Culture, Cell Separation, and Temperature-Stimulated Cell and Tissue Detachment

The TRPBCs are a promising group of materials for cell culture, cell separation, and tissue engineering. On the one hand, cell proliferation and tissue formation are strongly dependent on the physicochemical properties of the substrate with respect to the tissue, such as elasticity, hydrophobicity, and surface charge [126,127,128]. The required physicochemical properties of the substrate for cell culturing and tissue engineering can be adjusted by the chemical composition of the TRPBCs. On the other hand, the substrates should be biocompatible and highly stable (of low biodegradability) in physiological media. The biocompatibility of the TRPBCs is still under question today, since they either kill the cells (have a cytotoxic effect) or prevent cell adhesion (have a possible antifouling effect). Furthermore, polymers can degrade in physiological media or even during sterilization such that their functionality can be lost [129,130].

Substrates that are capable of temperature-controlled cells and tissue detachment are of great interest. TRPBCs are such promising materials. To engineer the TRPBCs for cell culture and temperature-stimulated detachment, the surface should provide the conditions for high cell adhesion, proliferation, and formation of the dense cellular biofilm in one state and should be capable of rapid cell or tissue detachment in the other state. Information about TRPBCs for cell culture, cell separation, and temperature-stimulated cell and tissue detachment is summarized in Table 4. The first attempts to use TRPBCs for tissue engineering were related to P*N*IPAM homopolymer grafted brushes [131,132,133]. After first successes, TRPBCs with various chemical structures and molecular designs were synthesized and characterized for applications in advanced biomedical applications. The grafted brush coatings with temperature and pH-responsive poly(*N*-methacryloyl-l-leucine) showed promising results for the cultivation of embryonic kidney cells (HEK 293) [54].

A similar approach was proposed by Takahashi and others [134], who manufactured the P*N*IPAM brush by RAFT polymerization using a dithiobenzoate-based chain transfer agent, which provided the conjugation with various functional groups. P*N*IPAM brush with the terminal carboxylic group (functionalized with 3-maleimidopropionic acid) showed high cell adhesion at T > LCST and rapid cell detachment at T < LCST. An interesting approach was realized in the works [92,135], in which inorganic nanoparticles (silver or calcium carbonate) were embedded in TRPBC, affecting cell adhesion and growth in a cell-dependent manner. In the work [92], the growth of keratinocyte HaCaT was compared for PDEGMA TRPBCs and PDEGMA TRPBCs with silver nanoparticles. It was observed that HaCaT cells grow faster on POEGMA TRPBCs with nanoparticles. In turn, for WM35 cancerous cells (melanoma), rather an opposite effect was observed. The number of cells was slightly smaller on the PDEGMA TRPBCs with silver nanoparticles, and after 144 h of incubation they only started to converge. On the contrary, on the “pure” polymer coating, the monocellular layer was already formed after the same incubation time. The comparison between the number of cells cultured for 24 h on PDEGMA TRPBCs with incorporated calcium carbonate nanoparticles and on “pure” PDEGMA TRPBCs showed an essential and slight reduction in adhesion for WM35 and HaCaT cell lines, respectively [135]. At this time, the completely anti-adhesive effect described for the osteoblastic cell line MC3T3-el on PDEGMA TRPBCs was absent and has been surpassed by the incorporation of nanoparticles. However, for longer culture times, the number of cells for both PDEGMA TRPBCs (i.e., “pure” and with embedded nanoparticles) was reduced by almost five times [135].

In the work [27], the adhesion and release of the model cells with temperature stimuli were examined for homopolymers, random and block copolymers based on *N*IPAM and 2-lactobionamidoethyl methacrylate. The fabricated brushes demonstrated selective adhesion of HepG2 cells at 37 °C and antifouling properties against NIH-3T3 fibroblasts. HepG2 cells detached from the random copolymer at 25 °C, while the block copolymer did not release the cells at these conditions. A mixture of different oligo(ethylene glycol) methacrylate)s was used in the works [136,137] to synthesize TRPBCs. At T = 37 °C, L929 mouse fibroblasts adhered efficiently and spread. At T < LCST a rapid cell rounding was observed, allowing cell detachment.

In the works [28,29], another approach to TRPBC engineering for cell thermal detachment was proposed. RGD was incorporated into poly(*N*IPAM)-b-poly(acrylic acid), increasing cell adhesion at 37 °C and not decreasing the ability to detach adhered cells by lowering the temperature below LCST [28]. A similar study was presented in [29] where TRPBC was built on the basis of PDEGMA. The incorporation of RGD increased the adhesion of 3T3 fibroblasts at 37 °C and PDEGMA provided temperature-responsive properties that allowed the release of the cells at 23 °C.

An interesting approach was demonstrated in the work [138], in which P*N*IPAM brushes were charged with the carboxylic group by copolymerizing the P*N*IPAM with 2-carboxyisopropylacrylamide. Cell adhesion was higher on the surface of the copolymer brush at T < LCST.

In the work [139], poly(*N*IPAM-*co*-*N*,*N*-dimethylaminopropylacrylamide -*co*-*N*-*tert*-butylacrylamide) brushes were used to fabricate the TRPBCs, that enabled the temperature-controlled adhesion and detachment of human bone marrow mesenchymal stem cells (hbmMSC). In contrast, other cells derived from human bone marrow (hbm-derived cells) did not adhere to the TRPBCs. Therefore, such TRPBCs can be used for mesenchymal stem cell separation. Figure 10 shows the adhesion of hbmMSC cells to the brushes at 37 °C and their detachment when the temperature was reduced below LCST to 20 °C. In turn, hbm-derived cells scarcely adhered on the TRPBC surface. The hbmMSC cells can be purified from other hbm-derived cells using TRBPCs and temperature variation.

The group of T. Okano developed a commercially available Nunc™ Multidishes with UpCell™ Surface [15,16] that allows cells to adhere from the culture medium, provides good conditions for cell sheet formation, and is capable of releasing the formed cell sheet by temperature stimuli. The device is built on TRPBC with P*N*IPAM, adhesive for cells at 37 °C, which becomes antifouling against cells at a temperature decreased below LCST, resulting in the release of the cell sheet formed. Typically, to detach cells and tissues from scaffolds, trypsin treatment is used. In contrast, the approach introduced by Okano’s group [15,16,140] does not require trypsin treatment. The concept of cell sheet engineering is sketched in Figure 11.

An analogous approach, applying TRPBCs with UCST, was presented in the work [38], where poly(*N*-acryloyl glycinamide-*co*-*N*-phenylacrylamide) TRPBCs with three different monomer ratios were synthesized to give tunable phase transition temperatures in solution. NIH-3T3 cells at 30 °C adhered to poly(*N*-acryloyl glycinamide-*co*-*N*-phenylacrylamide) brushes at 30 °C, below the UCST transition, and were released at 37 °C. Moreover, poly(cholesteryl methacrylate)-based coatings show strong potential for biomedical applications, which can be related to the liquid crystallite structure of this polymer and multiple temperature-induced transitions [67].

The application of TRPBCs with LCST or UCST for cell culturing, cell separation, and thermo-stimulated cell detachment is very promising for medicine and biotechnology. Undoubtedly, advanced TRPBCs should present novel properties compared to traditional P*N*IPAM coatings, including perfect biocompatibility, high adhesion, proliferation, and viability of the cells on the coatings, controlled and rapid cells or tissue detachment, appropriate transition temperature, and ability to separate different types of cells. Additionally, it is of great interest to apply in such studies TRPBCs with Tg that is close to physiological temperatures, such as poly(cholesteryl methacrylate) or PBMA, although information on their use is very limited at present.

### 5.3. TRPBCs for Temperature-Controlled Protein Adsorption

Protein adsorption is one of the key factors that determine the fate of materials under physiological conditions [141,142,143]. Numerous articles have recently reported the capability of TRPBCs for temperature-controlled adsorption of proteins [1,144,145,146]. Protein adsorption is preferably governed by nonspecific physical interactions, in which the surface charge and hydrophobicity of TRPBCs and proteins play a key role [147,148,149]. Hydrophobic surfaces interact with hydrophobic proteins via van der Waals or π-π interactions [150]. In turn, the hydrophilic surfaces interact with proteins that have a high charge [150]. Summarized information about the application of TRPBCs for temperature-controlled protein adsorption is presented in Table 5.

Strong bovine serum albumin (BSA) adsorption at T > LCST was shown in [151,152] for TRPBC based on the homopolymer P*N*IPAM compared to the essentially diminished adsorption of BSA at T < LCST. In a similar work [47], a significant change of lentil lectin adsorption to P*N*IPAM coatings was shown, from not visible at 20 °C to moderate at 26 °C and finally strong at 29 and 34 °C. Additionally, including carboxyl groups from the multifunctional initiator of polymerization in this system results in a decrease in thermal sensitivity with decreasing pH, leading to strong protein adsorption at all temperatures.

POEGMA-based TRPBCs demonstrate nonfouling properties, with no adsorbed proteins at all the studied temperatures: BSA adsorption was not observed for PDEGMA TRPBC [153]. Similar results were presented in the work [48], in which lentil lectin adsorption to POEGMA246 TRPBC was not observed at T > LCST as well as T < LCST. Carboxylic groups from the multifunctional initiator of polymerization in this system allowed us to switch protein adsorption from ultimately low at neutral and base pH to strong at acid pH.

Poly(4-vinylpyridine)-based TRPBCs showed significantly more efficient adsorption of BSA and human fibrinogen at T > LCST than at T < LCST [53]. Copolymer poly(4-vinylpyridine-*co*-OEGMA246) TRPBCs have demonstrated three-stage switching not only for surface wetting, but also for morphology and BSA adsorption [154] (Figure 12). In a similar work [104] a series of copolymer poly(4-vinylpyridine-*co*-DEGMA) brushes with temperature-switchable hydrophilic–hydrophobic balance and grafted to SiO_2_ nanoparticles was successfully synthesized and protein adsorption was studied using isothermal titration calorimetry (ITC). For individual proteins, such as human serum albumin (HSA), immunoglobulin G (IgG), and fibrinogen (Fbg); switchable high-/low-fouling properties were exhibited by the copolymer TRPBCs. The presence of 4-vinyl pyridine fragments in the copolymer favored the adsorption of proteins below and above LCST. DEGMA fragments, in turn, reduced (HSA) or entirely blocked (IgG and Fbg) adsorption.

TRPBCs with terpolymer brushes of poly(*N*IPAM-*co*- *N*,*N*-dimethylaminopropylacrylamide-*co*-*N*-*tert*-butylacrylamide) were fabricated and applied for the separation of HSA albumin and γ-globulin [155]. A negatively charged HSA was adsorbed on cationic terpolymer brush-modified silica beads at higher temperatures with a low concentration of phosphate buffer at pH = 7.0. In the work [156], TRPBCs with terpolymer brushes possessing the sulfonic acid group, poly(*N*IPAM-*co*-2-acrylamido-2-methylpropanesulfonic acid-*co*-*tert*-butylacrylamide were synthesized. The adsorption of basic proteins onto this TRPBC was promoted by an increase in temperature, and adsorbed proteins were successfully released by reducing the temperature. An exceptional application of copolymer TRPBCs grafted to silicon beads is presented in bioseparation chromatography [157]. T. Okano et al. presented the application of several types of TRPBCs for the purification of the solutions from various proteins.

In turn, for silica beads modified with poly(3-acrylamidopropyl trimethylammonium chloride)-*block*-P*N*IPAM TRPBCs, the elution of milk serum indicated that proteins α-lactalbumin and β-lactoglobulin adsorbed on the copolymer brush layer at high temperature and desorbed when the temperature was reduced [158]. Mixed TRPBCs based on poly(2-vinylpyridine) and P*N*IPAM were studied in the work [159]. The amount of proteins adsorbed was controlled depending on the composition and temperature of the environment. Similar results have been shown for mixed polymer brushes of cationic poly(*N*,*N*-dimethylaminopropyl acrylamide) and thermoresponsive P*N*IPAM homopolymer [160]. Protein mixtures, albumin, conalbumin, fibrinogen, and γ-globulin, can be separated simply by changing the temperature after adsorption on the mixed brush [160].

Another novel and perspective issue of temperature-regulated adsorption of proteins on TRPBC coatings is the control of protein orientations enabled by temperature. Using PBMA-based TRPCs with glass transition temperature Tg close to the physiological range, the group of A. Budkowski demonstrated temperature-controlled orientation of BSA and immunoglobulin G (IgG) [25], two proteins commonly applied in immunosensors or enzyme-linked immunosorbent assays. The different orientations of proteins, BSA and IgG, adsorbed to the TRPC below and above its Tg (Figure 13) were determined with time-of-flight secondary ion mass spectroscopy (ToF-SIMS), which combines the sensitivity to the outermost region of adsorbed proteins and to the composition of different protein domains [161]. The ToF-SIMS analysis also excluded any protein denaturation. The concluded change in the dominant orientation of the antibody from end-on to head-on for IgG molecules adsorbed at temperatures below and above Tg, respectively, was confirmed with binding assay by the decreasing amount of antigen bound to the pre-adsorbed IgG layers [25]. In contrast, the surface amount of IgG molecules increased with the adsorption temperature. In turn, BSA adsorbed to TRPC above Tg adopts a side-on orientation with the Albumin 3 domain exposed from the surface, whereas other domains are exposed at lower adsorption temperatures [25]. This might have induced the formation of surface-bound BSA dimers and led to the observed almost twofold increase in BSA adsorption for temperatures elevated from 10 °C to 35 °C [25]. Demonstrated temperature-controlled orientation of proteins on TRPBCs can be applied to obtain remote biorecognition control, e.g., for the binding of cell receptors or to fabricate switchable biosensing platforms.

The new generation of TRPBCs for temperature-controlled protein adsorption must satisfy some important criteria, including appropriate transition temperature (in the physiological ranges), as well as the ability to stimulate quick temperature-controlled protein adsorption/desorption, to enable protein separation, prevent protein denaturation, and orient protein molecules on the surface in a suitable manner.

## 6. Conclusions

Modern biomedical technologies predict the application of materials and devices that can not only play their role effectively but also enable remote control of their functions. One of the most prospective materials for these advanced biomedical applications are the materials based on TRPBCs. In this review, we focus mainly on TRPBCs for temperature-switchable bacteria killing, temperature-controlled protein adsorption, and cell culturing, as well as temperature-controlled adhesion/detachment of cells and tissues. Each desired application places some specific requirements on polymer coatings, and TRPBCs for advanced biomedical applications should meet the required criteria.

In this review, methods for the fabrication and characterization of TRPBCs were summarized, and the possibilities for their application, advantages and disadvantages were presented in detail. Special attention was paid to the mechanisms of the thermo-responsibility of the TRPBCs. Two main mechanisms responsible for transition in materials that found numerous applications, i.e., transition based on the critical solution temperature and the glass transition temperature, were described. It is well known that TRPBCs such as POEGMA and P*N*IPAM exhibiting LCST in the physiological range (10–37 °C) are a promising starting point for the fabrication of devices for advanced biomedical applications. Despite the popularity of these TRPBCs, their applications for advanced biomedical technologies are often limited because of serious barriers such as weak control over protein adsorption/desorption or cell adhesion/detachment, impact on the protein structure, etc. These issues can be solved using copolymer TRPBCs, post-synthesis modification of the grafted brushes, or inclusion of additional components in the structure of TRPBCs.

In this work, remarks are made on the criteria required for the fabrication of TRPBCs for each advanced biomedical application analyzed. Antibacterial TRPBCs must be highly effective against bacterial cells and nontoxic to eukaryotic cells, to present the self-cleaning effect with on/off switching by temperature, to have an appropriate transition temperature, and to provide long-term application including multiple cycles of temperature-induced transitions. As a rule, to fabricate the TRPBCs capable of temperature-stimulated bacteria killing, specific biologically-active substances (drugs) should be embedded into the coatings in appropriate amounts, providing high functionality of the drugs and not blocking the temperature response from the coatings. In the case of TRPBCs only capable of contact bacteria killing, their physicochemical properties should provide the conditions necessary for the adhesion of the bacteria. Moreover, drug release from TRPBCs should have a prolonged character to enable multiple functionalities. Above all, antibacterial TRPBCs oriented toward applications in biomedicine must be noncytotoxic.

The TRPBCs for cell culturing, cell separation, and thermo-stimulated cell detachment should provide the appropriate conditions for perfect biocompatibility, high adhesion, proliferation, and the viability of the cells on the coatings, controlled and rapid cell or tissue detachment, appropriate transition temperature, and the ability for separation of the different cellular types. In the literature, numerous papers report high cell adhesion and rapid release governed by the charged polymers. Moreover, the polymers applied for cell culturing and their thermo-stimulated release should be highly stable in the culture medium and tolerant to sterilization to preserve their functionality. One of the most promising directions is the application of TRPBCs for cell separation. Cell separation, also commonly referred to as cell isolation or cell sorting, is a process by which one or more specific cell populations are isolated from a heterogeneous mixture of cells. In the case of the TRPBCs it can be easily realized on the basis of different adhesions of various types of the cells on the coatings and their detachment induced by the temperature-modified physicochemical properties of TRPBCs. The application of TRPBCs for cell culture dishes that allow adhesion of cells from the culture solution at elevated temperatures and detachment of the whole cellular sheet at lower temperatures was described here, with the prominent example of the commercially available product, Nunc™ Multidishes with UpCell™ Surface.

Finally, lastly presented here in the application of TRPBCs is temperature-controlled nonspecific protein adsorption, governed by hydrophobic or polar physical interactions. Temperature-controlled protein adsorption depends on the properties of TRPBCs at different temperatures. The new generation of TRPBCs for temperature-controlled protein adsorption must satisfy some important criteria, including the appropriate transition temperature (in the physiological ranges), as well as the ability to stimulate quick temperature-controlled protein adsorption/desorption, to enable protein separation, to prevent protein denaturation, and to orient the protein molecules on the surface in a suitable manner. A completely new approach for temperature-controlled protein adsorption was described in the last part of the review. It enables tuning not only the amount of the adsorbed proteins but also their orientation and biological activity.

In the past, the first materials capable of interactions with bacterial and eukaryotic cells, tissues and proteins were intuitively chosen by scientists without ability to impact these objects in the controlled manner. The second generation of the materials for biomedical applications was essentially improved in comparison to the first one; the surfaces of these materials were often modified by substances that had no toxic effect on the objects studied. Very rarely have these materials had a controlled impact on the biological systems, which was realized mainly by tuning the chemical nature of the materials. At the present time, the new type of materials for biomedical applications with active remote impact on the biological object is developing and in some cases is included in medical protocols. The TRPBCs belong to these materials. Despite many advances, numerous challenges and opportunities in the field of TRPBCs for advanced biomedical technologies remain open.

Such fundamental issues as biocompatibility, high efficiency, selectivity of the action, stability, long-term and multiple uses, and temperature of the transition close to physiological temperatures (appropriate transition temperature) need to be resolved. Therefore, there is a constant need for new approaches to design surfaces that could meet all desired requirements.

The biocompatibility of the P*N*IPAM TRPBCs is cell-dependent and is not yet fully recognized. The potential cytotoxicity of new TRPBCs has not yet been investigated in detail. It seems that research on the biocompatibility of TRPBCs will grow rapidly in the next few years. Another important criterion is the high efficiency and selectivity of TRPBCs, which allow them to prevent unwanted processes and essentially reduce the time of biological reactions. The stability, long-term and multiple uses are related not only with the biocompatibility and ecological risks of the applications of the TRPBCs but also with the economic effects because fabrication and applications of the TRPBCs is still an expensive process. Additionally, it is worth mentioning that in many cases it is difficult to obtain the TRPBCs with the transition temperature close to physiological temperatures. Finally, multifunctional TRPBCs for advanced biomedical applications have great potential because biological reactions can be tuned by changing a few stimuli simultaneously, giving multiple advantages compared to traditional TRPBCs.

## Figures and Tables

**Figure 1 polymers-14-04245-f001:**
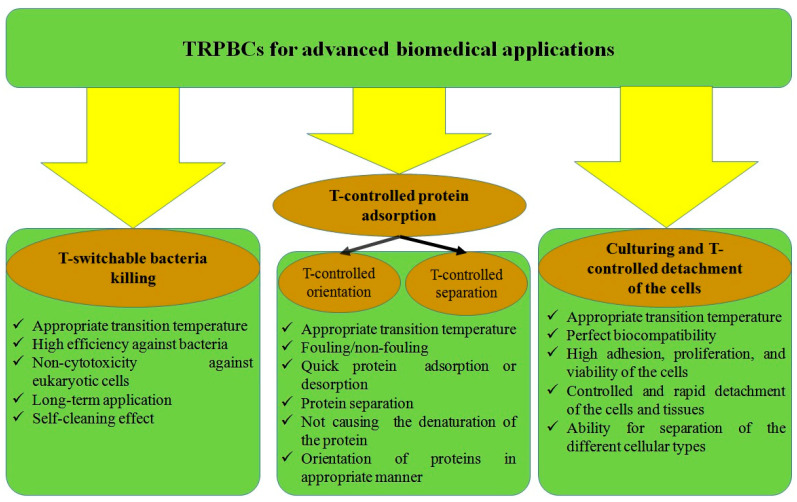
Advanced biomedical applications of the TRPBCs and the specific criteria required for these applications.

**Figure 2 polymers-14-04245-f002:**
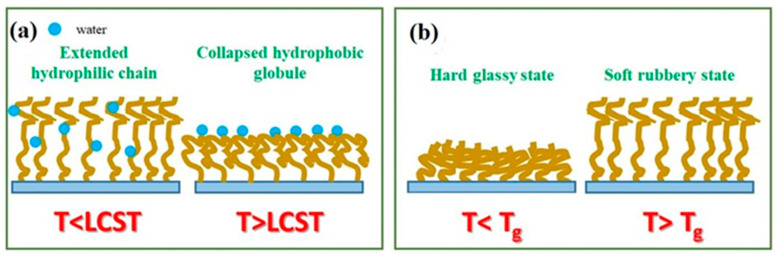
Schematic view of the transition of the TRPBCs from the extended hydrophilic chain to the collapsed hydrophobic globule caused by LCST (**a**) and the transition from the hard glassy state to the soft rubbery state (Tg) (**b**).

**Figure 3 polymers-14-04245-f003:**
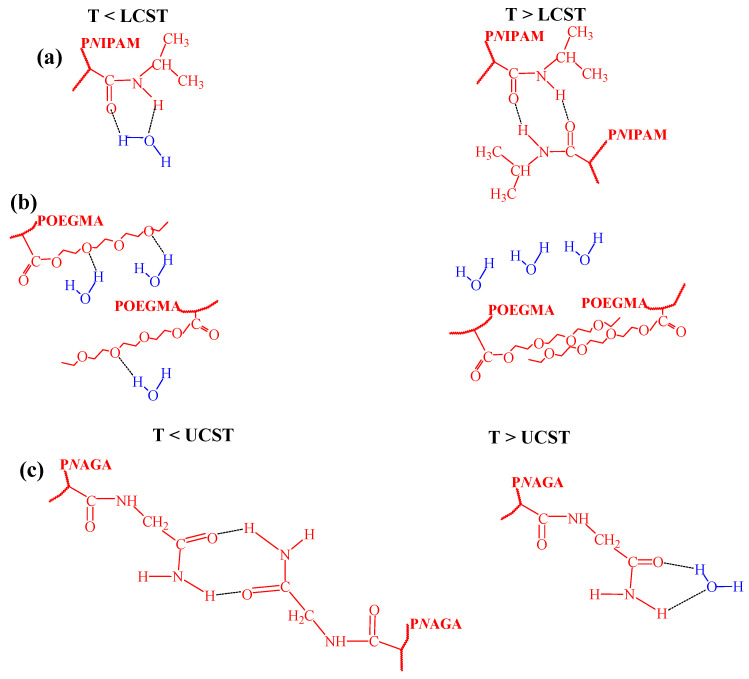
The simplified mechanisms of the LCST or UCST transitions for *N*IPAM (**a**), OEGMA (**b**) and *N*AGA (**c**) based TRPBCs.

**Figure 4 polymers-14-04245-f004:**
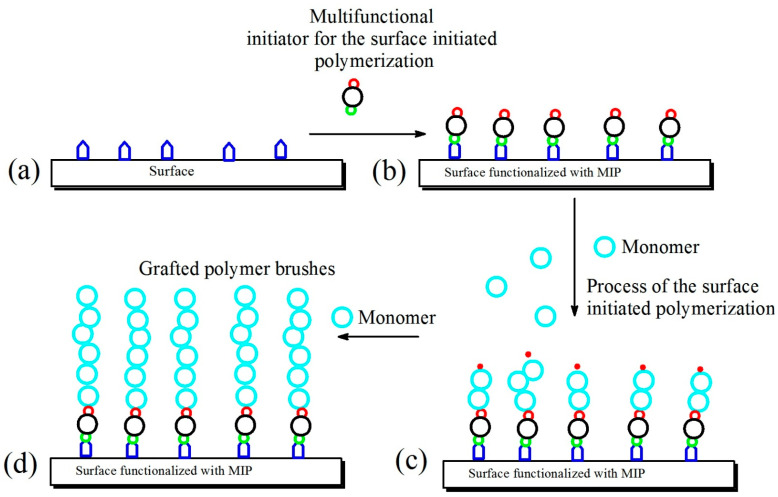
Functionalization of surfaces (**a**), subsequent grafting of a multifunctional initiator for a controlled living surface-initiated polymerization (**b**)**,** polymerization of a reactive monomer, initiated by reactive groups of the multifunctional initiator (**c**)**,** and the resulting grafted polymer brushes (**d**).

**Figure 5 polymers-14-04245-f005:**
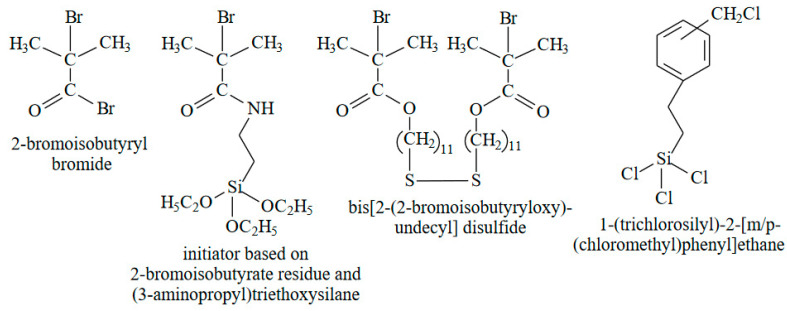
Typical multifunctional initiators for SI-ATRP.

**Figure 6 polymers-14-04245-f006:**
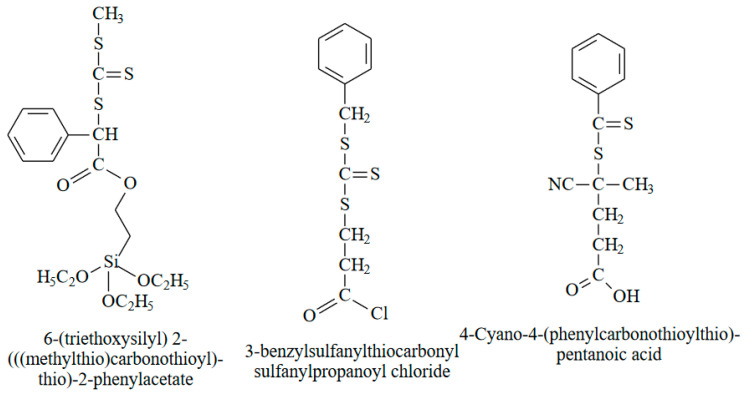
The most common multifunctional initiators for SI-RAFT.

**Figure 7 polymers-14-04245-f007:**
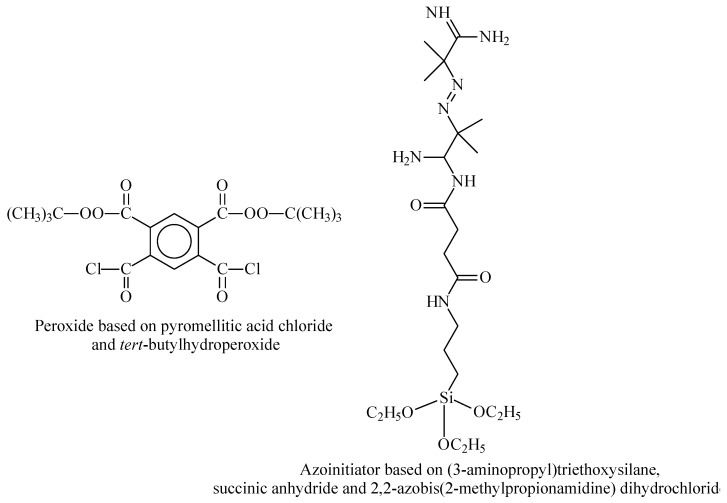
The most common multifunctional initiators for SI-PP or SI-AP.

**Figure 8 polymers-14-04245-f008:**
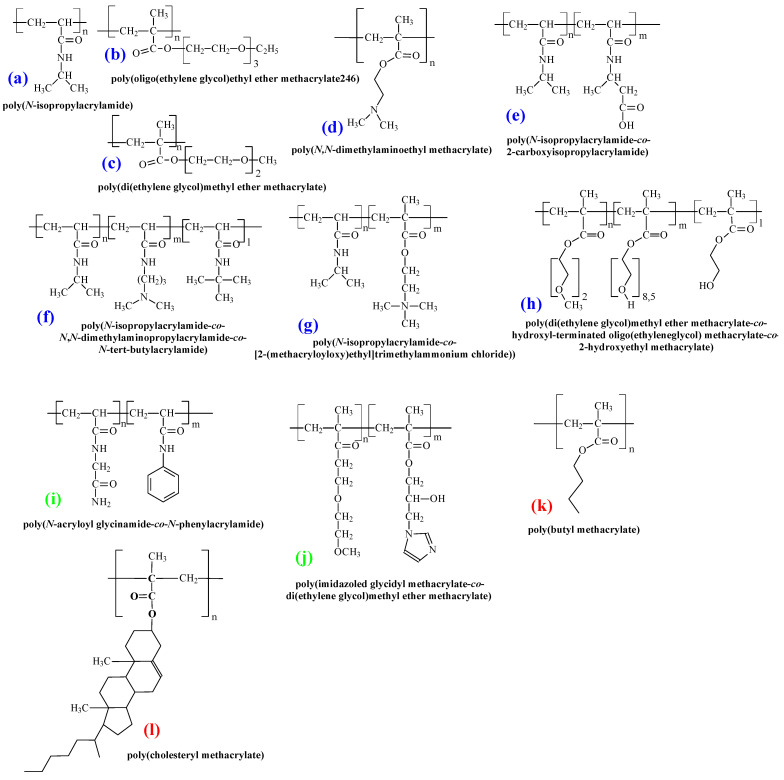
Chemical structures of TRPBCs for advanced biomedical applications. *N*-isopropylacrylamide (**a**), oligo(ethylene glycol)ethyl ether methacrylate with Mw = 246 (OEGMA246) (**b**), di(ethylene glycol)methyl ether methacrylate (**c**), *N*,*N*-dimethylaminoethyl methacrylate (**d**) and their copolymers with units from other functional monomers (**e**–**h**) poly(*N*-acryloyl glycinamide-*co*-*N*-phenylacrylamide) (**i**), poly(imidazoled glycidyl methacrylate-*co*-diethylene glycol methyl ether methacrylate) (**j**), poly(butyl methacrylate) (**k**) and poly(cholesteryl methacrylate) (**l**).

**Figure 9 polymers-14-04245-f009:**
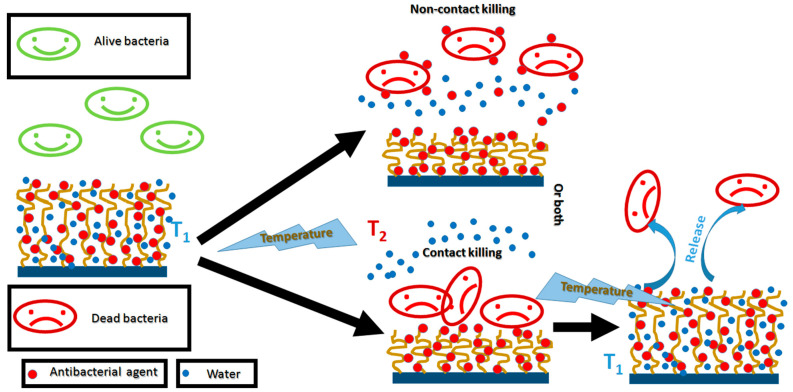
Controllable bacterial kill–release strategy based on TRPBCs.

**Figure 10 polymers-14-04245-f010:**
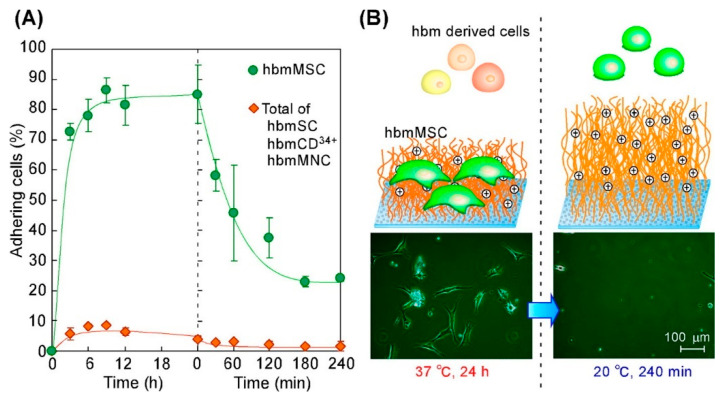
Adhesion and detachment profiles of human bone marrow mesenchymal stem cells (hbmMSC) and other human bone marrow-derived cells on poly(*N*IPAM-*co*-*N*,*N*-dimethylaminopropylacrylamide-*co*-*N*-*tert*-butylacrylamide) TRPBCs (**A**) and the mechanism of the separation of hbmMSC cells from other human bone marrow-derived cells (see text) (**B**) (with permission from [139]).

**Figure 11 polymers-14-04245-f011:**
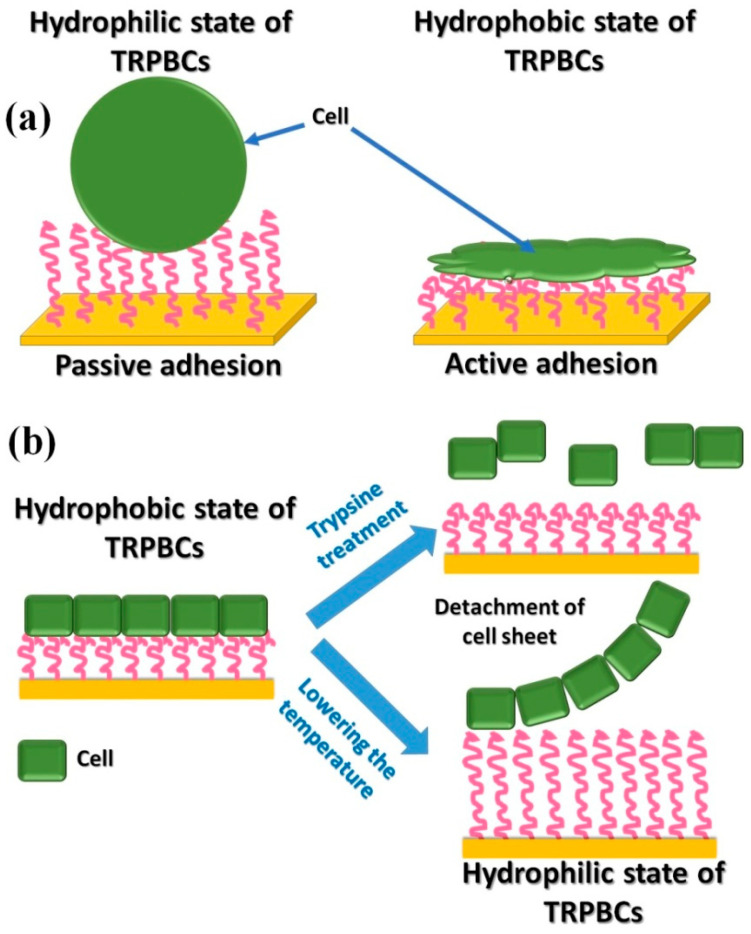
Interactions of TRPBCs with cells at T > LCST and T < LCST (**a**). The detachment of the cell sheet from the TRPBCs (**b**).

**Figure 12 polymers-14-04245-f012:**
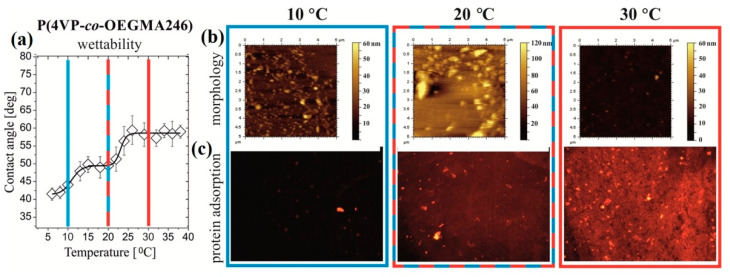
Temperature-controlled three-stage switching of wetting (**a**), morphology (**b**), and BSA absorption (**c**), determined for copolymer poly(4-vinylpyridine-*co*-OEGMA246) TRPBCs. Representative micrographs recorded with AFM (**b**) and fluorescence microscopy (for BSA molecules) labeled with Alexa Fluor (**c**) (with permission from [154]).

**Figure 13 polymers-14-04245-f013:**
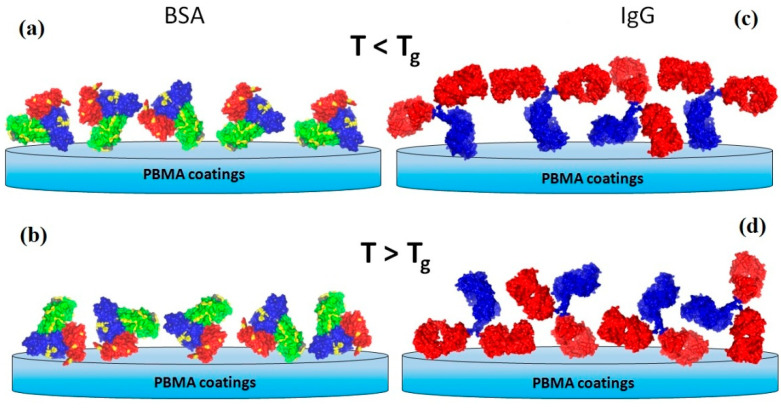
Orientation of the BSA and IgG proteins adsorbed to PBMA-based TRPC at a temperature below (**a**,**c**) and above (**b**,**d**) its glass transition (Tg). The different domains of BSA (red—Albumin 1, blue—Albumin 2, and green—Albumin 3) and IgG (red—Fab, blue—Fc) are distinguished by colors (modified with permission from [25]).

**Table 1 polymers-14-04245-t001:** The examples of multifunctional initiators for surface-initiated polymerizations.

Type of Polymerization Based on the Chemistry of Initiators	Multifunctional Initiators
SI-ATRP	1-(trichlorosilyl)-2-[m/p-(chloromethyl)phenyl]ethane [75]
	2-(4-chlorosulfonylphenyl)ethylsilane [76]
	2-bromoisobutyrate residues [77]
SI-RAFT	3-benzylsulfanylthiocarbonyl sulfanylpropanoyl chloride [84]
SI-PET-RAFT	2-(dodecylthiocarbonothioylthio)-2-methylpropionic acid [86]
	4-cyano-4-(phenylcarbonothioylthio)pentanoic acid [87]
	2-(n-butyltrithiocarbonate) propionic acid [87]
SI-PP	peroxide based on pyromellitic acid chloride and tert-butylhydroperoxide [88]
	cholesterol-based peroxide [89]
SI-AP	2,2-azobis(2-methylpropionamidine) dihydrochloride [90]
	asymmetric azobisisobutyronitrile-based trichlorosilane initiator [91]

**Table 2 polymers-14-04245-t002:** Methods for characterization of the TRPBCs and determination of their transition temperatures.

Method	Type of the Surface with TRPBCs	Determination of LCST	Determination of Tg
Measurement of the wetting contact angles	Flat and Curved	The temperature dependence of water contact angles is similar to that of a sigmoid with the deflection point at LCST [1,47,48,92,93,94,95]	Non-applicable [25,31]
Ellipsometry	Flat	The swelling ratio of TRPBCs decreases sharply at the LCST [96,97,98]	The thermal expansion curve contains the deflection point at the glass transition temperature [70,71,72]
Atomic force microscopy (AFM)	Flat	LCST affects surface morphology, but defining LCST is almost impossible [49]	The RMS roughness decreases above the glass transition temperature [25,31]
Dynamic light scattering (DLS)	Dispersive	The hydrodynamic radius of the nano-object decreases above LCST twice at least [64,99,100,101]	It is difficult to determine the glass transition by measuring the hydrodynamic radius of the particles, but approaches for the detection of the glass transition with DLS are proposed [102,103,104]
Differential scanning calorimetry (DSC)	Dispersive	The DSC thermogram contains the endothermic peak at the LCST [105]	The DSC heating curve contains the deflection at the glass transition temperature [106,107,108]
Turbidity measurements	Dispersive and Flat	Above the LCST, the TRPBCs are more turbid than below [109,110]	Non-applicable

**Table 3 polymers-14-04245-t003:** Summarized information about antibacterial TRPBCs.

Type of the Polymer Brushes, Polymerization Technique and References	Antibacterial Agents	Comments
**Without Antibacterial Agents**		
**Homopolymer Grafted Brushes**		
P*N*IPAM, SI-AP, argon plasma polymerization [115,116,117,118,119]	None	The adhesion and detachment of bacterial cells depend on the physicochemical properties of bacterial surfaces and TRPBCs. Adhesion of *Cobetia marina* (*Staphylococcus epidermidis*) at T > LCST and release (rinsing) at T < LCST. *Salmonella typhimurium* and *Bacillus cereus* strong adhesion at T > LCST and weak adhesion at T < LCST
**Copolymer Grafted Brushes**		
Poly(*N*IPAM-*co*-acrylamide) (85 to 15 mol%), SI-AP [119]	None	*Salmonella typhimurium* and *Bacillus cereus* almost same adhesion at different T
poly(*N*IPAM-*co*-*N*-*tert*-butylacrylamide) (80 to 20 mol%), SI-AP [119]		*Salmonella typhimurium* and *Bacillus cereus* very strong adhesion at T > LCST and very weak adhesion at T < LCST
**With Antibacterial Agents**		
**Homopolymer Grafted Brushes**		
*PDEGMA*, SI-ATRP [120]	Levofloxacin	*Staphylococcus aureus* was tested No traces of bacterial biofilm at T > LCST
*PDEGMA*, SI-ATRP [92]	Silver nanoparticles	*E. coli* and *S. aureus* were killed at T > LCST
Poly(4-vinylpyridine), SI-ATRP [92,93]	Silver nanoparticles	*E. coli* and *S. aureus* were killed at T > LCST
**Copolymer Grafted Brushes**		
Poly(*N*IPAM-*co*-[2-(methacryloyloxy)-ethyl]trimethylammonium chloride), SI-RAFT, photoinduced polymerization from double bonds [121,122]	[2-(methacryloyloxy)ethyl]trimethylammonium chloride	*E. coli* and *S. aureus* were killed at T > LCST
		Detachment of the dead bacteria at T < LCST; no detachment at T > LCST
Poly(*N*IPAM-*co*-2-carboxyethyl acrylate) modified by vancomycin moieties, SI-PIMP [124]	Vancomycin	*E. coli* and *S. aureus* were killed at T < LCST
		Detachment of the bacteria at T > LCST
Poly(DEGMA-*co*-hydroxyl-terminated oligo(ethylene glycol) methacrylate-*co*-2-hydroxyethyl methacrylate), SI-ATRP [125]	Magainin I peptide	*L. ivanovii* and *E. coli* were preferably killed at T < LCST
		Detachment of the dead bacteria at T > LCST
**Mixed Grafted Brushes**		
P*N*IPAM and poly [2-(methacryloyloxy)ethyl]-trimethylammonium chloride, ATRP and then assembled onto surface [123]	Poly [2-(methacryloyloxy) ethyl]trimethylammonium chloride	*S. aureus* was killed at T > LCST
		Detachment of the dead bacteria at T < LCST

**Table 4 polymers-14-04245-t004:** Summarized information about TRPBCs for cell culture, cell separation, and temperature-stimulated cell and tissue detachment.

Type of the Polymer Brushes, Polymerization Technique and References	Application
**TRPBCs with LCST**	
**Homopolymer Grafted Brushes**	
P*N*IPAM, electron beam polymerization [131,132,133]	At 37 °C adherent for the different types of cells, once the temperature is decreased, the TRPBCs become antifouling against the cells and the formed cellular sheet releases.
Poly(*N*-methacryloyl-*l*-leucine), SI-PP [54]	The cultivation of embryonic kidney cell (HEK 293)
**Homopolymer Grafted Brushes Functionalized with End Groups**	
P*N*IPAM brushes with the terminal carboxylic group (functionalized with 3-maleimidopropionic acid), SI-RAFT [134]	High cell adhesion at the temperature above the LCST and rapid cell detachment at the temperature below LCST
**Homopolymer Grafted Brushes with Nanoparticles**	
PDEGMA brushes with embedded inorganic nanoparticles, SI-ATRP [92,135]	Modification of the properties of TRPBCs by inorganic nanoparticles. Keratinocyte HaCaT grows faster on the PDEGMA TRPBCs with silver nanoparticles than on the PDEGMA TRPBCs. Cancerous cells WM35 (melanoma) grow slightly slower on PDEGMA TRPBCs with silver nanoparticles than on PDEGMA TRPBCs. The comparison between the number of cells cultured 24 h on PDEGMA TRPBCs with incorporated calcium carbonate nanoparticles and on “pure” PDEGMA TRPBCs shows an essential and slight reduction in adhesion for the WM35 and HaCaT cell lines, respectively. The completely anti-adhesive effect described for the osteoblastic cell line MC3T3-el on PDEGMA TRPBCs was absent and has been surpassed by the incorporation of nanoparticles. For longer culture times, the number of cells for both PDEGMA TRPBCs (i.e., “pure” and with embedded nanoparticles) was reduced by almost five times
**Copolymer Grafted Brushes**	
**Random**	
Poly(DEGMA-*co*-oligo(ethylene glycol) methacrylate), SI-ATRP [136,137]	L929 mouse fibroblasts at T = 37 °C adhered efficiently and spread well. At T < LCST a rapid cell rounding was observed allowing cells to detach
Poly(*N*IPAM-*co*-2-lactobionamidoethyl methacrylate), SI-ATRP [27]	Selective adhesion of HepG2 cells at T = 37 °C and antifouling properties against NIH-3T3 fibroblasts. HepG2 cells detached at 25 °C
Poly(*N*IPAM-*co*-2-carboxyisopropylacrylamide), electron beam polymerization [138]	Cell adhesion was higher on the surface of copolymer brushes at T < LCST
PDEGMA with RGD peptide, SI-ATRP [29]	Incorporation of RGD increased adhesion of 3T3 fibroblasts at T = 37 °C; the cells released at T < LCST
Poly(*N*IPAM-*co*-*N*,*N*-dimethylaminopropylacrylamide-*co*-*N*-*tert*-butylacrylamide), SI-ATRP [139]	Human bone marrow mesenchymal stem cells (hbmMSC) adhered to the brushes at 37 °C and were detached below LCST at 20 °C. Other bone marrow-derived cells (hbmMSC) did not adhere to the brushes. Hence, the brushes can be used to purify hbmMSC cells from the hbm-derived cells
**Block Copolymers**	
Poly(*N*IPAM)-*block*-poly(acrylic acid) with RGD peptide, SI-ATRP [28]	The RGD increased the adhesion of the cells at 37 °C and did not decrease the ability to detach the adhered cells by lowering the temperature below LCST
**TRPBCs with UCST**	
Poly(*N*-acryloyl glycinamide-*co*-*N*-phenylacrylamide), SI-ATRP [38]	NIH-3T3 cells adhered at 30 °C, which is below the UCST transition, and were released at 37 °C
**TRPBCs with Tg**	
Poly(cholesteryl methacylate), SI-PP [67]	Culture of non-malignant bladder cancer cells (HCV29 line) and granulosa cells

**Table 5 polymers-14-04245-t005:** Summarized information about application of the TRPBCs for temperature-controlled protein adsorption.

Type of the Polymer Brushes, Polymerization Technique and References	Application
**TRPBCs with LCST**	
**Homopolymer Grafted Brushes**	
P*N*IPAM, SI-ATRP [151,152]	Strong BSA adsorption at T > LCST. Low BSA adsorption at T < LCST
P*N*IPAM with carboxylic groups from multifunctional initiator, SI-PP [47]	Strong adsorption of lentil lectin at T > LCST. Low lentil lectin adsorption at T < LCST. Strong protein adsorption for all T at acid pH
PDEGMA, SI-ATRP [153]	Non-fouling properties observed for BSA
POEGMA246 with carboxylic groups from multifunctional initiator, SI-PP [48]	Non-fouling properties (for lentin lectin) for all T at neutral and base pH. Strong lentin lectin adsorption for all T at acid pH
Poly(4-vinylpyridine), SI-PP [53]	More efficient BSA and human fibrinogen adsorption at T > LCST than at T < LCST
**Copolymer Grafted Brushes**	
Poly(4-vinylpyridine-*co*-OEGMA246), SI-PP [154] Poly(4-vinylpyridine-*co*-DEGMA), SI-ATRP [104]	Three-stage switching in BSA adsorption Switchable high/low fouling properties for human serum albumin, immunoglobulin G and fibrinogen
Poly(*N*IPAM-*co*-*N*,*N*-dimethylaminopropylacrylamide-*co*-*N*-*tert*-butylacrylamide), SI-ATRP [155]	For separation of human serum albumin and γ-globulin Human serum albumin adsorbed at T > LCST
Poly(*N*IPAM-*co*-2-acrylamido-2-methylpropanesulfonic acid-*co*-*tert*-butylacrylamide, SI-ATRP [156]	The adsorption of basic proteins is promoted by elevating the temperature. Adsorbed proteins released by reducing the temperature
**Block Copolymers**	
Poly(3-acrylamidopropyl trimethylammonium chloride)-*block*-P*N*IPAM, SI-ATRP [158]	α-lactalbumin and β-lactoglobulin from milk adsorbed at T > LCST and desorbed at T < LCST
Mixed Grafted Brushes	
Poly(2-vinylpyridine) and P*N*IPAM, grafted using monocarboxy-terminated polymers [159]	The amount of protein adsorbed could be controlled, depending on composition and the temperature
Poly(*N*,*N*-dimethylaminopropyl acrylamide) and P*N*IPAM, SI-RAFT [160]	Protein mixtures, albumin, conalbumin, fibrinogen, and γ-globulin, can be separated simply by changing the temperature after adsorption on the mixed brush
**TRPBCs with Tg**	
PBMA, SI-ATRP [25]	Almost twice the increase in BSA adsorption for the temperature elevated from 10 °C to 35 °C. Temperature-dependent BSA orientation, with Albumin 1 and 2 (Albumin 3) exposed for the protein adsorbed at temperature below (above) Tg. The adsorption of IgG increased with temperature. Temperature-dependent IgG orientation, with end-on (head-on) alignment for the protein adsorbed at temperature below (above) Tg

## Data Availability

Not applicable.

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
