# Peer review of "Temperature-Responsive Polymer Brush Coatings for Advanced Biomedical Applications"

_polymers, 2022, doi:10.3390/polym14194245_

Round 1

Reviewer 1 Report

The author in this review introduced a comprehensive study about the utilization of temperature-responsive polymer brush coatings (TRPBCs) in biomedical applications like temperature-switchable bacteria killing, temperature-controlled protein adsorption, culturing of cells, and temperature-controlled adhesion/detachment of cells and tissues are considered. In addition to the fabrication, characterization, advantages, and disadvantages of TRPBCs and I have some questions below:

1-    There are many other key parameters that could affect the TRPBCs. We know that the author focuses on the transitions based on LCST and Tg, but it is worthy to explain the effect of other factors on TRPBCs like UCST, Surface and Volume Hydrophilicity of Hydrophilic Polymer Brushes, and contact angle briefly in the part of “Mechanisms of the temperature-induced transition of TRPBCs” “Page 4 and 5”.

Two example articles as follows:

1-    Zhuang, P.; Dirani, A.; Glinel, K.; Jonas, A. M. Temperature dependence of the surface and volume hydrophilicity of hydrophilic polymer brushes. Langmuir 2016, 32 (14), 3433-3444.

2-    lemming, P.; Fery, A.; Münch, A. S.; Uhlmann, P. Does Chain Confinement Affect Thermoresponsiveness? A Comparative Study of the LCST and Induced UCST Transition of Tailored Grafting-to Polyelectrolyte Brushes. Macromolecules 2022, 55 (15), 6775-6786.

·      In this review, the author explained and illustrated the applications of the Temperature-responsive polymer brush coatings only in four figures, such as Figures 10, 11, 12, and 13. It is better if the author includes more articles to provide a wider explanation of the applications.

·      In the introduction (page 2), the author mentioned that the surface-initiated (SI) polymerization techniques used for the synthesis of polymer brush, such as “surface-initiated atom transfer radical polymerization (SI-ATRP), surface reversible addition fragmentation chain transfer polymerization (SI-RAFT), surface-initiated nitroxide-mediated polymerization (SI-NMP), surface-initiated photoiniferter-mediated polymerization (SI-PIMP), surface-initiated photopolymerization (SI-PhotoP) and surface-initiated polymerization using peroxide initiators or azo initiators (SI-PP or SI-AP).”

Also, Surface-Initiated Ring-Opening Metathesis Polymerization was used as a fundamental technology for preparing polymer brushes. please see the following articles.

1-    Ye, Q.; He, B.; Zhang, Y.; Zhang, J.; Liu, S.; Zhou, F. Grafting robust thick zwitterionic polymer brushes via subsurface-initiated ring-opening metathesis polymerization for antimicrobial and anti-biofouling. ACS applied materials & interfaces 2019, 11 (42), 39171-39178.

2-    Chen, Q. Surface-Initiated Ring-Opening Metathesis Polymerization (SI-ROMP): History, General Features, and Applications in Surface Engineering with Polymer Brushes. International Journal of Polymer Science 2021, 2021.

· On pages 8 and 9, the author explained the known steps of SI-ATRP and SI-RAFT generally, but he didn’t cite some articles used in the brush polymer coating.

Also, SI-PET-RAFT is considered a more advanced technology which should be mentioned in that part. After discussing the examples of SI-ATRP, SI-RAFT, and SI-PET-RAFT, it may be better to collect their photoinitiators in one table and also cite these initiators.

·      The figures need more polishing. The font size should be clear in figure 12, with decreasing the figure size in figures 11 and 13. Also, we have pale figures like figure 10, which should be introduced in high resolution. All figures should be modified to be matched together in font size ……etc.

·      Tables 3 and 4 summarize the applications of the TRPBCs for temperature-controlled protein adsorption. The tables only contain the polymer`s name and application. It is worthy of providing a column for the grafting methodology (polymerization technique).

·      It would be good to include a future outlook of the TRPBCs in the conclusion section.

Other mistakes:

1-    Put “s” to the word “multiple use” in page 1

2-    In page 2: “is enables” should be “is enabled”

3-    In page 4: Put “,” after “Below the LCST”

4-    In page 4: “This affects also the morphology of” should be “This also affects the morphology of”.

5-    In Page 5: Put “,” before “and refractive index”

6-    Put “s” to all biomedical application

7-    In page 10: put “,” after “Usually”

8-    In page 12: Put “,” before “and methods specific for” and “and binding assays”

9-    In page 12: Put “,” after “Contrary to LCST”

Author Response

Responses to the Reviewer #1 comments:

  1. There are many other key parameters that could affect the TRPBCs. We know that the author focuses on the transitions based on LCST and Tg, but it is worthy to explain the effect of other factors on TRPBCs like UCST, Surface and Volume Hydrophilicity of Hydrophilic Polymer Brushes, and contact angle briefly in the part of “Mechanisms of the temperature-induced transition of TRPBCs” “Page 4 and 5”.

Two example articles as follows:

1-    Zhuang, P.; Dirani, A.; Glinel, K.; Jonas, A. M. Temperature dependence of the surface and volume hydrophilicity of hydrophilic polymer brushes. Langmuir 2016, 32 (14), 3433-3444.

2-    Flemming, P.; Fery, A.; Münch, A. S.; Uhlmann, P. Does Chain Confinement Affect Thermoresponsiveness? A Comparative Study of the LCST and Induced UCST Transition of Tailored Grafting-to Polyelectrolyte Brushes. Macromolecules 2022, 55 (15), 6775-6786.

Reply. Appropriate discussion was added. The recommended references were mentioned in the main text.

“The crucial parameters determining the properties of the thermoresponsive grafted polymer brushes include polymer density, molecular weight, topology, and chemical component dissolved in the solvent or included in the structure of the macromolecules. The behavior of poly(N-isopropylacrylamide) (PNIPAM) grafted brushes at low grafting densities and molecular weights, as well as at high grafting densities and molecular weights, was described by Leckband et al. [58]. At low densities of grafting and molecular weights, the formation of lateral aggregates or “octopus micelles” was demonstrated. In contrast, at high grafting densities and molecular weights, the PNIPAM-grafted brush coatings collapsed uniformly. In turn, Benetti [59-60] showed the impact of macromolecular topology (linear versus cyclic polymer brushes) on colloidal stability and bio-inertness as well as temperature responsivity. It was noted that the properties and behavior of the examined polymer brushes were significantly different, even at the same temperature.

Another interesting work [61] reported the synthesis and detailed characterization of thermoresponsive poly(N,N-dimethylaminoethyl methacrylate) TRPBC with well-controlled molecular weight and grafting density (0.08−0.20 chains/nm2). For this material, a well-pronounced LCST transition is observed with a reduction in brush layer thickness of more than 40% by spectroscopic ellipsometry at intermediate grafting densities (0.12−0.20 chains/nm2) in 5 mM NaCl solution. In turn, the UCST transition, induced by multivalent [Fe(CN)6]3− ions, reaches a remarkable change in layer thickness of ∼80% already at the lowest investigated grafting density of 0.08 chains/nm2.” 

“Transitions are accompanied by changes in the volume and surface hydrophilicity. In the work [51] a series of dense water-swollen polymer brushes was studied using contact angle measurements, ellipsometry and quartz crystal microbalance. Diagrams of surface versus volume hydrophilicity of the brushes allow one to identify two types of behavior: strongly water-swollen brushes exhibit a progressive decrease of volume hydrophilicity with temperature, while surface hydrophilicity changes moderately; weakly water-swollen brushes have a close-to-constant volume hydrophilicity, while surface hydrophilicity decreases with temperature. Thermoresponsive brushes abruptly switch from one behavior to the other and do not exhibit an abrupt change of surface hydrophilicity throughout their collapse transition. In general, there is no direct correlation between surface hydrophilicity and volume hydrophilicity, because surface properties depend on the details of conformation and composition at the surface, while volume properties are averaged over a finite region within the brush [51]. In contrast to results reported in [51], our works ([1] and references therein) always showed strong changes in surface hydrophilicity at temperature-induced transitions. These differences may be related to different structures of TRPBCs (thickness, grafting density, or other factors).”

2. In this review, the author explained and illustrated the applications of the Temperature-responsive polymer brush coatings only in four figures, such as Figures 10, 11, 12, and 13. It is better if the author includes more articles to provide a wider explanation of the applications.

 Reply. We took attention to the most critical challenges that can be solved using temperature-responsive polymer brush coatings and gave to paper the most amazing figures to illustrate the problem. In our opinion, the presented figures are entirely enough; more amount of figures can confuse the Readers diverting them from the essence of the matter.

3. In the introduction (page 2), the author mentioned that the surface-initiated (SI) polymerization techniques used for the synthesis of polymer brush, such as “surface-initiated atom transfer radical polymerization (SI-ATRP), surface reversible addition fragmentation chain transfer polymerization (SI-RAFT), surface-initiated nitroxide-mediated polymerization (SI-NMP), surface-initiated photoiniferter-mediated polymerization (SI-PIMP), surface-initiated photopolymerization (SI-PhotoP) and surface-initiated polymerization using peroxide initiators or azo initiators (SI-PP or SI-AP).”

Also, Surface-Initiated Ring-Opening Metathesis Polymerization was used as a fundamental technology for preparing polymer brushes. please see the following articles.

  • Ye, Q.; He, B.; Zhang, Y.; Zhang, J.; Liu, S.; Zhou, F. Grafting robust thick zwitterionic polymer brushes via subsurface-initiated ring-opening metathesis polymerization for antimicrobial and anti-biofouling. ACS applied materials & interfaces 2019, 11 (42), 39171-39178.
  • Chen, Q. Surface-Initiated Ring-Opening Metathesis Polymerization (SI-ROMP): History, General Features, and Applications in Surface Engineering with Polymer Brushes. International Journal of Polymer Science 2021, 2021.

Reply. Appropriate discussion was added. The recommended references were mentioned in the main text.

In addition, surface-initiated ring-opening metathesis polymerization (SI-ROMP) was used as a fundamental technology for the preparation of polymer brushes. SI-ROMP offers an effective way of polymerization of norbornene monomers, allowing the preparation of specific polymers that are not accessible by other polymerization methods [73-74].

4. On pages 8 and 9, the author explained the known steps of SI-ATRP and SI-RAFT generally, but he didn’t cite some articles used in the brush polymer coating.

Also, SI-PET-RAFT is considered a more advanced technology which should be mentioned in that part. After discussing the examples of SI-ATRP, SI-RAFT, and SI-PET-RAFT, it may be better to collect their photoinitiators in one table and also cite these initiators.

 Reply. Appropriate discussion and Table were added in the main text.

“Special attention should be paid to the more progressive type of SI-RAFT named as surface-initiated photoinduced electron transfer-reversible addition–fragmentation chain transfer polymerization (SI-PET-RAFT). This method allows surface functionalization with spatiotemporal control and provides oxygen tolerance under ambient conditions [86-87]. The modularity and versatility of SI-PET-RAFT are highlighted through significant flexibility with respect to the choice of monomer, light source and wavelength, and photoredox catalyst. The ability to obtain complex patterns in the presence of air is a significant advantage compared to other controllable surface-initiated polymerization methods [86-87].”

In Table 1 examples of the most common multifunctional initiators for surface initiated polymerizations are summarized.

Table 1. The examples of multifunctional initiators for surface initiated polymerizations.

Type of polymerization based on the chemistry of initiators

Multifunctional initiators

SI-ATRP

1-(trichlorosilyl)-2-[m/p-(chloromethyl)phenyl]ethane [75]

2-(4-chlorosulfonylphenyl)ethylsilane [76]

2-bromoisobutyrate residues [77]

SI-RAFT

3-benzylsulfanylthiocarbonyl sulfanylpropanoyl chloride [84]

SI-PET-RAFT

2-(dodecylthiocarbonothioylthio)-2-methylpropionic acid [86]

4-cyano-4-(phenylcarbonothioylthio)pentanoic acid [87]

2-(n-butyltrithiocarbonate) propionic acid [87]

SI-PP

peroxide based on pyromellitic acid chloride

and tert-butylhydroperoxide [88]

cholesterol-based peroxide [89]

SI-AP

2,2-azobis(2-methylpropionamidine) dihydrochloride [90]

asymmetric azobisisobutyronitrile‐based trichlorosilane initiator [91]

5. The figures need more polishing. The font size should be clear in figure 12, with decreasing the figure size in figures 11 and 13. Also, we have pale figures like figure 10, which should be introduced in high resolution. All figures should be modified to be matched together in font size ……etc

Reply. The figures in high resolution have been introduced in the manuscript.

 6. Tables 3 and 4 summarize the applications of the TRPBCs for temperature-controlled protein adsorption. The tables only contain the polymer`s name and application. It is worthy of providing a column for the grafting methodology (polymerization technique).

Reply. Information about grafting methodology was introduced in the Tables.

 7. It would be good to include a future outlook of the TRPBCs in the conclusion section.

Reply. A future outlook of the TRPBCs was included in the conclusion section.

In the past, the first materials capable of interactions with bacterial and eukaryotic cells, tissues and proteins were intuitively chosen by scientists without ability to impact these objects in the controlled manner. The second generation of the materials for biomedical applications was essentially improved in comparison to the first one; the surfaces of these materials were often modified by substances that had no toxic effect on the objects studied. Very rarely have these materials had a controlled impact on the biological systems, which was realized mainly by tuning the chemical nature of the materials. At the present time, the new type of materials for biomedical applications with active remote impact on the biological object is developing and in some cases is included in medical protocols. The TRPBCs belong to these materials. Despite many advances, numerous challenges and opportunities in the field of TRPBCs for advanced biomedical technologies remain open.

Such fundamental issues as biocompatibility, high efficiency, selectivity of the action, stability, long-term and multiple uses, and temperature of the transition close to physiological temperatures (appropriate transition temperature) need to be resolved. Therefore, there is a constant need for new approaches to design surfaces that could meet all desired requirements.

The biocompatibility of the PNIPAM TRPBCs is cell-dependent and is not yet fully recognized. The potential cytotoxicity of new TRPBCs has not yet been investigated in detail. It seems that the research on the biocompatibility of TRPBCs will grow rapidly in the next few years. Another important criterion is the high efficiency and selectivity of TRPBCs, which allow them to prevent unwanted processes and essentially reduce the time of biological reactions. The stability, long-term and multiple uses are related not only with the biocompatibility and ecological risks of the applications of the TRPBCs but also with the economic effects because fabrication and applications of the TRPBCs is still an expensive process. Additionally, it is worth mentioning that in many cases it is difficult to obtain the TRPBCs with the transition temperature close to physiological temperatures. Finally, multifunctional TRPBCs for advanced biomedical applications have great potential because biological reactions can be tuned by changing a few stimuli simultaneously, giving multiple advantages compared to traditional TRPBCs.” 

 8. Other mistakes:

  • Put “s” to the word “multiple use” in page 1
  • In page 2: “is enables” should be “is enabled”
  • In page 4: Put “,” after “Below the LCST”
  • In page 4: “This affects also the morphology of” should be “This also affects the morphology of”.
  • In Page 5: Put “,” before “and refractive index”
  • Put “s” to all biomedical application
  • In page 10: put “,” after “Usually”
  • In page 12: Put “,” before “and methods specific for” and “and binding assays”
  • In page 12: Put “,” after “Contrary to LCST”

Reply. All text was carefully checked, and mistakes and typos were corrected.

Reviewer 2 Report

This paper discussed the temperature-responsive polymer brush coatings from the synthesis, mechanism and applications. Detailed information and previous studies have been discussed within the draft. Overall, it is a good review paper for this are. However, there are still some problems within this draft:

  1. The authors covered detailed information of this topic. However, it seems the flow of the discussion not perfectly organized. As shown in in Figure 1. three main approaches for the advanced biomedical applications were listed. I was expecting the authors would structure the article accordingly. It seems the order for 3.2 and 3.3 is not exactly the same as the Figure 1. In addition, the literature discussed within section 3 could be organized according to the each bullet point mentioned within Figure 1. It would make the structure perfect!
  2. In addition, for section 2, the subtitle is “Mechanisms of the temperature-induced transition of TRPBCs”. Section 2.1 and 2.2 fit the topic very well while section 2.3 and 2.4 are more related to the fabrication and characterization. I do not think they really fit the topic. I would recommend either change the subtitle of section 2, or make another section for 2.3 and 2.4 since both of them are very important.
  3. LCST and UCST are discussed in detail within the draft. I would recommend to explain the fundamental mechanisms of these two behaviors to fully complete the discussion. Current discussion only covers the phenomena can be observed (polymer chain collapse or extend at different temperature. A short explanation for the mechanism would be helpful.
  4. Between line 278 and 279, the authors indicated the polymer chain growth only happens on R-group (R-group approach), which is not accurate. Z-group can also be the initiation group for the polymer chain growth (Z-group approach). https://doi.org/10.1002/pola.23032;  https://doi.org/10.1002/marc.200600223
  5. Between line 343 and 344, the authors mentioned water contact angle can only be measured at a flat surface, which is not accurate. Water contact angle method can also be applied on curved surface: https://doi.org/10.1016/j.jcis.2011.08.019
  6. Between line 402 and 403, the authors claimed “The hydrophobic nanoparticle is surrounded by a hydrophobic hydration water layer that does not form hydrogen bonds.” I might be wrong but I do not think this is the right information delivered by the paper. Please comment on that.
  7. Between 477 and 479, the authors claimed “Quaternary ammonium salts effectively killed the attached bacteria (E. coli and S. aureus) at T>LCST and the dead bacteria were detached by reducing the temperature below the LCST. No detachment was observed at T>LCST.” However, I do not think this is correct according to the original paper. According to the original paper, the bacteria will be captured at a temperature higher than LCST and released at a temperature lower than LCST.
  8. It seems only a few figures have the permissions (Figure 10, Figure12 and Figure 13). Please comment if the authors made other figures by themselves. If not, the permissions should be obtained before being published.
  9. Polymer brushes are very attractive for their special structures but the fabrication process typically is difficult. There are coming studies of using bottlebrush polymers as a replacement for polymer brushes, particularly bottlebrush polymer for surface modifications. A short discussion with that should be discussed and references should be cited: https://doi.org/10.1039/C5CS00579E; https://doi.org/10.1021/acsmacrolett.0c00384; https://doi.org/10.1039/C8SM01127C; https://doi.org/10.1021/acs.macromol.9b01801; https://doi.org/10.1021/acs.macromol.0c00744 ; https://doi.org/10.1021/acs.langmuir.0c01675; https://doi.org/10.1016/j.actbio.2017.01.061 
  10. There is a typo within line 162. It should be “PNIPAM” instead of “PNIMAM”

Author Response

Responses to the Reviewer #2 comments:

  1. The authors covered detailed information of this topic. However, it seems the flow of the discussion not perfectly organized. As shown in in Figure 1. three main approaches for the advanced biomedical applications were listed. I was expecting the authors would structure the article accordingly. It seems the order for 3.2 and 3.3 is not exactly the same as the Figure 1. In addition, the literature discussed within section 3 could be organized according to the each bullet point mentioned within Figure 1. It would make the structure perfect!

Reply. Figure 1 is drawn from an aesthetic point of view, where information about TRPBCs for protein adsorption and orientation is placed in the center to achieve symmetry. The structure of the manuscript is related to its importance and amazing applications. In our opinion, the most interesting and actual are antibacterial applications, then tissue engineering, and finally protein adsorption and orientation. We can not organize the literature discussed within section 3 to each bullet point mentioned in Figure 1 because this topic is completely new and the specific criteria required for these applications are the product of our analysis.                                                                                         

2. In addition, for section 2, the subtitle is “Mechanisms of the temperature-induced transition of TRPBCs”. Section 2.1 and 2.2 fit the topic very well while section 2.3 and 2.4 are more related to the fabrication and characterization. I do not think they really fit the topic. I would recommend either change the subtitle of section 2, or make another section for 2.3 and 2.4 since both of them are very important.

 Reply. The enumeration of the subtitles was changed as recommended by the Reviewer.

 3. LCST and UCST are discussed in detail within the draft. I would recommend to explain the fundamental mechanisms of these two behaviors to fully complete the discussion. Current discussion only covers the phenomena can be observed (polymer chain collapse or extend at different temperature. A short explanation for the mechanism would be helpful.

 Reply. Appropriate discussion was added

In polymer chemistry, the phenomenon of the LCST is related to the systems based on polymer-solvent mixtures that are miscible below a given critical temperature and turn to two-phase unmixed systems above this critical temperature. The Gibbs free energy change (ΔG) related to the mixing of these two phases is negative below the LCST and positive above it, and the entropy change ΔS = – (dΔG/dT) is negative for the mixing process. This is in contrast to the more common and intuitive case, in which the entropy change promotes mixing because of the increased volume accessible to each component upon mixing. In general, the unfavorable entropy of mixing responsible for the LCST may have two physical origins. The first is related to interactions between the two components, such as strong polar interactions or hydrogen bonds, which prevent random mixing. The second physical factor that can lead to LCST is compressibility effects, especially in polymer-solvent systems [43-44]. In contrast to the LCST, the UCST is the critical temperature above which the components of a mixture are miscible in all proportions. Phase separation at the UCST is driven by unfavorable energetics; in particular, interactions between components favor a partially demixed state [43-44].”

 4. Between line 278 and 279, the authors indicated the polymer chain growth only happens on R-group (R-group approach), which is not accurate. Z-group can also be the initiation group for the polymer chain growth (Z-group approach). https://doi.org/10.1002/pola.23032;  https://doi.org/10.1002/marc.200600223

 Reply. Appropriate changes were done. The recommended references were included.

Attachment of the RAFT agent through the Z-group or the R-group can considerably influence the outcome of polymer brush grafting [84-85]. As a rule, the R-group initiates the growth of the majority of polymer chains, and the Z-group stabilizes the intermediate radical species. In this case, R-designed RAFT agents allow the termination of two macroradicals on the surface, resulting in the loss of RAFT agent. In contrast, Z-designed RAFT agents prevent these side reactions. However, the transfer of the macroradical to the RAFT agent takes place close to the surface. With increasing brush length, the thiocarbonylthio group may become less and less accessible and the growing polymer brush has a shielding effect [84-85].”

5. Between line 343 and 344, the authors mentioned water contact angle can only be measured at a flat surface, which is not accurate. Water contact angle method can also be applied on curved surface: https://doi.org/10.1016/j.jcis.2011.08.019

 Reply. Appropriate discussion was added.

“The measurements of the contact angles are not applicable for brushes that are immobilized on the dispersive surfaces. In fact, drop shape visualization and contact angle measurement are also possible on curved surfaces [95] but these measurements were not performed for TRPBCs [25,31].”

 6. Between line 402 and 403, the authors claimed “The hydrophobic nanoparticle is surrounded by a hydrophobic hydration water layer that does not form hydrogen bonds.” I might be wrong but I do not think this is the right information delivered by the paper. Please comment on that.

 Reply. Information was presented in a more suitable manner.

The hydrophobic nanoparticles are surrounded by “secondary bound water” that performs hydrophobic hydration around the side chain carbon atoms, where cage-like water formations around these carbons allow the polymer to remain in the water [49, 104].”

 7. Between 477 and 479, the authors claimed “Quaternary ammonium salts effectively killed the attached bacteria (E. coli and S. aureus) at T>LCST and the dead bacteria were detached by reducing the temperature below the LCST. No detachment was observed at T>LCST.” However, I do not think this is correct according to the original paper. According to the original paper, the bacteria will be captured at a temperature higher than LCST and released at a temperature lower than LCST.

 Reply. We have checked this information once again, presented information is correct.

 8. It seems only a few figures have the permissions (Figure 10, Figure12 and Figure 13). Please comment if the authors made other figures by themselves. If not, the permissions should be obtained before being published.

 Reply. The authors made other figures by themselves.

 9. Polymer brushes are very attractive for their special structures but the fabrication process typically is difficult. There are coming studies of using bottlebrush polymers as a replacement for polymer brushes, particularly bottlebrush polymer for surface modifications. A short discussion with that should be discussed and references should be cited: https://doi.org/10.1039/C5CS00579E; https://doi.org/10.1021/acsmacrolett.0c00384; https://doi.org/10.1039/C8SM01127C; https://doi.org/10.1021/acs.macromol.9b01801; https://doi.org/10.1021/acs.macromol.0c00744 ; https://doi.org/10.1021/acs.langmuir.0c01675; https://doi.org/10.1016/j.actbio.2017.01.061

 Reply. Appropriate discussion was added, recommended references were cited.

A similar class of materials, bottlebrush polymers, have also been actively developed in the last years [6]. Bottlebrush polymers have densely grafted side chains along the linear backbones with extended cylindrical shapes with no entanglements. Smart bottlebrush polymers can be used for the fabrication of well-organized and predictable coatings on the solid surfaces [7-14], but, in contrary to grafted polymer brushes, their long-term applications in the physiological environment and in contact with biological objects are questionable.”

 10. There is a typo within line 162. It should be “PNIPAM” instead of “PNIMAM”

Reply. All text was carefully checked, and mistakes and typos were corrected.

Round 2

Reviewer 2 Report

The authors have addressed all my questions/comments.